# AN ADDITIVE INSTANCE-WISE APPROACH TO MULTI-CLASS MODEL INTERPRETATION

**Vy Vo**[1]   **Van Nguyen**[1]   **Trung Le**[1]
**Quan Hung Tran**[2]   **Gholamreza Haffari**[1]   **Seyit Camtepe**[3]   **Dinh Phung**[1,4]

[1]Monash University, Australia
[2]Adobe Research, USA
[3]CSIRO's Data61, Australia
[4]VinAI Research, Vietnam

## ABSTRACT

Interpretable machine learning offers insights into what factors drive a certain prediction of a black-box system. A large number of interpreting methods focus on identifying explanatory input features, which generally fall into two main categories: attribution and selection. A popular attribution-based approach is to exploit local neighborhoods for learning instance-specific explainers in an additive manner. The process is thus inefficient and susceptible to poorly-conditioned samples. Meanwhile, many selection-based methods directly optimize local feature distributions in an instance-wise training framework, thereby being capable of leveraging global information from other inputs. However, they can only interpret single-class predictions and many suffer from inconsistency across different settings, due to a strict reliance on a pre-defined number of features selected. This work exploits the strengths of both methods and proposes a framework for learning local explanations simultaneously for multiple target classes. Our model explainer significantly outperforms additive and instance-wise counterparts on faithfulness with more compact and comprehensible explanations. We also demonstrate the capacity to select stable and important features through extensive experiments on various data sets and black-box model architectures.

## 1 INTRODUCTION

Black-box machine learning systems enjoy a remarkable predictive performance at the cost of interpretability. This trade-off has motivated a number of interpreting approaches for explaining the behavior of these complex models. Such explanations are particularly useful for high-stakes applications such as healthcare (Caruana et al., 2015; Rich, 2016), cybersecurity (Nguyen et al., 2021) or criminal investigation (Lipton, 2018). While model interpretation can be done in various ways (Mothilal et al., 2020; Bodria et al., 2021), our discussion will focus on feature importance or saliency-based approach - that is, to assign relative importance weights to individual features w.r.t the model's prediction on an input example. Features here refer to input components interpretable to humans; for high-dimensional data such as texts or images, features can be a bag of words/phrases or a group of pixels/super-pixels (Ribeiro et al., 2016). Explanations are generally made by selecting top $K$ features with the highest weights, signifying $K$ most important features to a black-box's decision. Note that this work tackles feature selection locally for an input data point, instead of generating global explanations for an entire dataset.

An abundance of interpreting works follows the removal-based explanation approach (Covert et al., 2021), which quantifies the importance of features by removing them from the model. Based on how feature influence is summarized into an explanation, methods in this line of works can be broadly categorized as *feature attribution* and *feature selection*. In general, attribution methods produce relative importance scores to each feature, whereas selection methods directly identify the subset of features most relevant to the model behavior being explained. One popular approach to learn attribution is through an **Additive** model (Ribeiro et al., 2016; Zafar & Khan, 2019; Zhao et al., 2021). The underlying principle is originally proposed by LIME (Ribeiro et al., 2016) which

learns a regularized linear model for each input example wherein each coefficient represents feature importance scores. LIME explainer takes the form of a linear model $w.z$ where $z$ denotes neighboring examples sampled heuristically around the input [1]. Though highly interpretable themselves, additive methods are inefficient since they optimize individual explainers for every input. As opposed to the instance-specific nature of the additive model, most of the feature selection methods are developed instance-wisely (Chen et al., 2018; Bang et al., 2021; Yoon et al., 2019; Jethani et al., 2021a). **Instance-wise** frameworks entail global training of a model approximating the local distributions over subsets of input features. Post-hoc explanations can thus be obtained simultaneously for multiple instances.

**Contributions.** In this work, we propose a novel strategy integrating both approaches into an **additive instance-wise** framework that simultaneously tackles all issues discussed above. The framework consists of 2 main components: an explainer and a feature selector. The explainer first learns the local attributions of features across the space of the response variable via a multi-class explanation module denoted as $W$. This module interacts with the input vector in an additive manner forming a linear classifier locally approximating the black-box decision. To support the learning of local explanations, the feature selector constructs local distributions that can generate high-quality neighboring samples on which the explainer can be trained effectively. Both components are jointly optimized via backpropagation. Unlike such works as (Chen et al., 2018; Bang et al., 2021) that are sensitive to the choice of $K$ as a hyper-parameter, our learning process eliminates this reliance (See Appendix G for a detailed analysis on why this is necessary).

Our contributions are summarized as follows

- We introduce **AIM** - an **A**dditive **I**nstance-wise approach to **M**ulti-class model interpretation. Our model explainer inherits merits from both families of methods: model-agnosticism, flexibility while supporting efficient interpretation for multiple decision classes. To the best of our knowledge, we are the first to integrate *additive* and *instance-wise* approaches into an end-to-end amortized framework that produces such a multi-class explanation facility.

- Our model explainer is shown to produce remarkably faithful explanations of high quality and compactness. Through quantitative and human assessment results, we achieve superior performance over the baselines on different datasets and architectures of the black-box model.

## 2 RELATED WORK

Early interpreting methods are gradient-based in which gradient values are used to estimate attribution scores, which quantifies how much a change in an input feature affects the black-box's prediction in infinitesimal regions around the input. It originally involves back-propagation for calculating the gradients of the output neuron w.r.t the input features (Simonyan et al., 2014). This early approach however suffers from vanishing gradients during the backward pass through ReLU layers that can downgrade important features. Several methods are proposed to improve the propagation rule (Bach et al., 2015; Springenberg et al., 2014; Shrikumar et al., 2017; Sundararajan et al., 2017). Since explanations based on raw gradients tend to be noisy highlighting meaningless variations, a refined approach is sampling-based gradient, in which sampling is done according to a prior distribution for computing the gradients of probability Baehrens et al. (2010) or expectation function (Smilkov et al., 2017; Adebayo et al., 2018). Functional Information (FI) (Gat et al., 2022) is the state-of-the-art in this line of research applying functional entropy to compute feature attributions. FI is shown to work on auditory, visual and textual modalities, whereas most of the previous gradient-based methods are solely applicable to images.

A burgeoning body of works in recent years can be broadly categorized as removal-based explanation (Covert et al., 2021). Common removal techniques include replacing feature values with neutral or user-defined values such as zero or Gaussian noises (Zeiler & Fergus, 2014; Dabkowski & Gal, 2017; Fong & Vedaldi, 2017; Petsiuk et al., 2018; Fong et al., 2019), marginalization of distributions over input features (Lundberg & Lee, 2017; Covert et al., 2020; Datta et al., 2016), or substituting held-out feature values with samples from the same distribution (Ribeiro et al., 2016). The output explanations are often either attribution-based or selection-based. In addition to additive models

---

[1] $z$ is a binary representation vector of an input indicating the presence/absence of features. The dot-product operation is equivalent to summing up feature weights given by the weight vector $w$, giving rise to additivity.

that estimate feature important via the coefficients of the linear model, feature attributions can be calculated using Shapley values (Datta et al., 2016; Lundberg & Lee, 2017; Covert et al., 2020) or directly obtained by measuring the changes in the predictive probabilities or prediction losses when adding or excluding certain features (Zeiler & Fergus, 2014; Schwab & Karlen, 2019). On the other hand, selection-based works straightforwardly determine which subset of features are important or unimportant to the model behavior under analysis (Chen et al., 2018; Yoon et al., 2019; Bang et al., 2021; Jethani et al., 2021a; Nguyen et al., 2021; 2022). Explanations are made by either selecting features with the highest logit scores obtained from the learned feature distribution, or specifying a threshold between 0 and 1 to decide on the most probable features. Most selection methods adopt amortized optimization, thus post-hoc inference of features for multiple inputs can be done very efficiently. In contrast, attribution-based approaches are mostly less efficient since they process input examples individually. There have been methods focusing on improving the computational cost of these models (Dabkowski & Gal, 2017; Schwab & Karlen, 2019; Jethani et al., 2021b).

Recently, there is an emerging interest in integrating the *instance-wise* property into an *additive* framework to better exploit global information. For a given input, Plumb et al. (2018); Yoon et al. (2022) in particular learn a surrogate model assigning weights to training examples such that those more similar or relevant to the input are given higher weights. A locally interpretable model (that is often LIME-based) is subsequently trained on these samples to return feature attributions. Agarwal et al. (2021) further proposes a neural additive framework that constructs an explainer in a form of a linear combination of neural networks. Despite their potential, these methods have only been reported to work on tabular data.

## 3 PROPOSED METHOD

### 3.1 PROBLEM SETUP

In the scope of this paper, we limit the current discussion to classification problems. Consider a data set of pairs $(X, Y)$ where $X \sim \mathbb{P}_X(\cdot)$ is the input random variable and $Y$ is characterized by the conditional distribution $\mathbb{P}_m(Y \mid X)$ obtained as the predictions of a pre-trained black-box classifier for the response variable. The notation $m$ stands for *model*, indicating the predictive distribution of the black-box model, to be differentiated from the ground-truth distribution. We denote $\boldsymbol{x} \in \mathbb{R}^d$ as an input realization with $d$ interpretable features and predicted label $Y = c \in \{1, ..., C\}$. Given an input $\boldsymbol{x}$, we obtain the hard prediction from the black-box model as $y_m = \mathrm{argmax}_c \, \mathbb{P}_m(Y = c \mid X = \boldsymbol{x})$.

While earlier methods generate a single $d-$dimensional weight vector $\boldsymbol{w_x}$ assigning the importance weights to each feature, we define an **explainer** $\mathcal{E} : \mathbb{R}^d \mapsto \mathbb{R}^{d \times C}$ mapping an input $\boldsymbol{x}$ to a weight matrix $\boldsymbol{W_x} \in \mathbb{R}^{d \times C}$ with the entry $W_{\boldsymbol{x}}^{i,j}$ representing the relative weights of the $i$th feature of $\boldsymbol{x}$ to the predicted label $j \in \{1, ..., C\}$.

$$\mathcal{E}(\boldsymbol{x}) = \boldsymbol{W_x}.$$

Given a training batch, LIME (Ribeiro et al., 2016) in particular trains separate explainers for every input, thus cannot take advantage of the global information from the entire dataset. In line with the *instance-wise* motivation, our explainer $\mathcal{E}$ is trained globally over all training examples to produce local explanations with respect to individual inputs simultaneously, which also seeks to effectively enable global behavior (e.g., two similar instances should have similar explanations). As $\mathcal{E}$ is expected to be locally faithful to the black-box model, we optimize $\mathcal{E}$ on the local neighborhood around the input $\boldsymbol{x}$. This region is constructed via a **feature selection** module. We now explain how this is done.

### 3.2 TRAINING OBJECTIVES

Let $\mathbf{z} \in \{0, 1\}^d$ be a random variable with the entry $z^i = 1$ indicating the feature $i$th is important to the black-box's predictions. With respect to $\boldsymbol{x}$, we employ a **selector** $\mathcal{S} : \mathbb{R}^d \mapsto [0, 1]^d$ that outputs $\mathcal{S}(\boldsymbol{x}) = \pi_{\boldsymbol{x}}$ such that $\pi_{\boldsymbol{x}}^i := \mathbb{P}(z^i = 1 \mid X = \boldsymbol{x}), i = 1, ..., d$.

Through the probability vector $\pi_{\boldsymbol{x}}$, the selector helps define a local distribution on a local space of samples $\boldsymbol{z_x} \odot \boldsymbol{x}$ with $\boldsymbol{z_x} \sim \mathrm{MultiBernoulli}(\pi_{\boldsymbol{x}})$ and element-wise product $\odot$. The selector $\mathcal{S}$ is also a learnable module, and we want it to generate well-behaved local samples that focus more on valuable features/attributions of $\boldsymbol{x}$. Intuitively, if the feature $i$ of $\boldsymbol{x}$ contributes more to the predictions of the black-box model, i.e., $\pi_{\boldsymbol{x}}^i \approx 1$, the explainer is expected to give higher assignments to the row

vector $W_{\boldsymbol{x}}^{i,:}$. To mimic how the black-box model behaves towards different attributions, we propose to minimize the cross-entropy loss between the prediction of the black-box model on local examples $\boldsymbol{z_x} \odot \boldsymbol{x}$ and the prediction of the explainer on binary vectors $\boldsymbol{z_x}$ via the weight matrix $\boldsymbol{W_x}$ as

$$\mathcal{L}_1 = \mathbb{E}_{\boldsymbol{x}}\mathbb{E}_{\boldsymbol{z_x}}\Big[\text{CE}\left(\tilde{y}_m, \text{softmax}(\boldsymbol{W_x}^T \boldsymbol{z_x})\right)\Big], \tag{1}$$

where CE is the cross-entropy function and $\tilde{y}_m = \text{argmax}_c\, \mathbb{P}_m(Y = c \mid \boldsymbol{z_x} \odot \boldsymbol{x})$.

To make the process continuous and differentiable for training, the temperature-dependent Gumbel-Softmax trick (Jang et al., 2016; Maddison et al., 2016) is applied for relaxing Bernoulli variables $z_{\boldsymbol{x}}^i$. In particular, the continuous representation $\tilde{z}_{\boldsymbol{x}}^i$ is sampled from the Concrete distribution as $\left[\tilde{z}_{\boldsymbol{x}}^i, 1 - \tilde{z}_{\boldsymbol{x}}^i\right] \sim \text{Concrete}(\pi_{\boldsymbol{x}}^i, 1 - \pi_{\boldsymbol{x}}^i)$:

$$\tilde{z}_{\boldsymbol{x}}^i = \frac{\exp\{\left(\log \pi_{\boldsymbol{x}}^i + G_{i1}\right)/\tau\}}{\exp\{(\log(1 - \pi_{\boldsymbol{x}}^i) + G_{i0})/\tau\} + \exp\{(\log \pi_{\boldsymbol{x}}^i + G_{i1})/\tau)\}},$$

with temperature $\tau$, random noises $G_{i0}$ and $G_{i1}$ independently drawn from **Gumbel** distribution $G_t = -\log(-\log u_t),\ u_t \sim \textbf{Uniform}(0, 1)$.

Given the corresponding prediction $\tilde{y}_m = \text{argmax}_c\, \mathbb{P}_m(Y = c \mid \tilde{\boldsymbol{z}}_{\boldsymbol{x}} \odot \boldsymbol{x})$, $\mathcal{L}_1$ now becomes

$$\mathcal{L}_1 = \mathbb{E}_{\boldsymbol{x}}\mathbb{E}_{\tilde{\boldsymbol{z}}_{\boldsymbol{x}}}\Big[\text{CE}\left(\tilde{y}_m, \text{softmax}(\boldsymbol{W_x}^T \tilde{\boldsymbol{z}}_{\boldsymbol{x}})\right)\Big]. \tag{2}$$

Since $\boldsymbol{z_x}$ is a binary vector indicating the absence/presence of features, $\boldsymbol{z_x} \odot \boldsymbol{x}$ indeed acts as a local perturbation, which generally concurs with the principle of LIME model. However, different from LIME, we amortize the explainer $\mathcal{E}$ to produce the weight matrices $\boldsymbol{W_x}$ locally approximating the black-box model with linear classifiers operating on input neighborhoods. Furthermore, we replace LIME's uniform sampling strategy with a learnable local distribution offered by the selector $\mathcal{S}$.

We argue that heuristic sampling is inadequate for our purpose. As $d$ gets large, realizing the space of $2^d$ possible binary patterns is infeasible. Given the fact that the number of binary patterns that actually approximate the original prediction is arbitrarily small, it is thus very difficult for such a simple linear separator as one used in LIME to learn useful patterns within finite sampling rounds. While diversity in these samples is desirable for learning attributions for individual decision classes, we also want the explainer $\mathcal{E}$ to focus more on relevant features to the original prediction $y_m$. To encourage the selector to yield more of the samples that contain the features that best approximate the model behavior on the original input, we propose the following information-theoretic approach.

Let $\boldsymbol{x}_{\mathbb{S}}$ denote the sub-vector formed by the subset of $K$ most important features $\mathbb{S} = \{i_1, \ldots, i_K\} \subset \{1, \ldots, d\}$ $(i_i < i_2 < \cdots < i_K)$. Thus, $\pi_{\boldsymbol{x}}^i$ can now be viewed as the probability that the $i$th feature of $\boldsymbol{x}$ appears in $\mathbb{S}$. Given a random vector $X_{\mathbb{S}} \in \mathbb{R}^K$, we maximize the mutual information.

$$\mathbb{I}(X_{\mathbb{S}}; Y) = \mathbb{E}\Big[\log \frac{\mathbb{P}_m(Y \mid X_{\mathbb{S}})}{\mathbb{P}_m(Y)}\Big] = \mathbb{E}_X \mathbb{E}_{\mathbb{S}|X}\mathbb{E}_{Y|X_{\mathbb{S}}}\Big[\log \mathbb{P}_m(Y \mid X_{\mathbb{S}})\Big] + \text{Constant}. \tag{3}$$

Based on the following inequality, we can obtain a variational lower bound for $\mathbb{I}(X_{\mathbb{S}}; Y)$ via a generic choice of conditional distribution $\mathbb{Q}_{\mathbb{S}}(Y \mid X_{\mathbb{S}})$

$$\mathbb{E}_{Y|X_{\mathbb{S}}}[\log \mathbb{P}_m(Y \mid X_{\mathbb{S}})] = \mathbb{E}_{Y|X_{\mathbb{S}}}[\log \mathbb{Q}_{\mathbb{S}}(Y \mid X_{\mathbb{S}})] + \text{KL}(\mathbb{P}_m(Y \mid X_{\mathbb{S}}), \mathbb{Q}_{\mathbb{S}}(Y \mid X_{\mathbb{S}}))$$
$$\geq \mathbb{E}_{Y|X_{\mathbb{S}}}[\log \mathbb{Q}_{\mathbb{S}}(Y \mid X_{\mathbb{S}})],$$

where KL represents the Kullback-Leibler divergence.

It is worth noting that the purpose of using the mutual information in L2X (Chen et al., 2018) and our AIM framework are different. L2X uses the mutual information to directly select valuable features/attributions. Meanwhile, in our work, the role of the selector is to balance exploration with exploitation. Stochastic sampling yields various examples that produce different predictions, and $\mathcal{E}$ exploits such variation to learn feature attributions w.r.t multiple classes. Simultaneously, maximizing $\mathbb{I}(X_{\mathbb{S}}; Y)$ encourages the selector $\mathcal{S}$ to produce a meaningful probability vector focusing more on

the selected subset of attributions that can well approximate the full-input decision. In Appendix D, we show that an explainer with learnable local distributions performs significantly better than one optimized on heuristic examples.

Maximizing the mutual information in Eq. (3) can therefore be relaxed to maximizing the variational lower bound $\mathbb{E}_X \mathbb{E}_{\mathbb{S}|X} \mathbb{E}_{Y|X_\mathbb{S}} \Big[ \log \mathbb{Q}_\mathbb{S}(Y \mid X_\mathbb{S}) \Big]$. We parametrize $\mathbb{Q}$ with a function approximator $\mathcal{G}$ such that $\mathbb{Q}_\mathbb{S}(\boldsymbol{x}_\mathbb{S}) := \mathcal{G}(\boldsymbol{x}_\mathbb{S})$. Notice that we can now use the element-wise product $\tilde{\boldsymbol{z}}_{\boldsymbol{x}} \odot \boldsymbol{x}$ to approximate $\boldsymbol{x}_\mathbb{S}$. If $\boldsymbol{x}$ contains discrete features (e.g., words), we embed a feature (e.g., a selected word) in $\mathbb{S}$ with a learnable embedding vector, wherein a feature not in $\mathbb{S}$ is replaced with a zero vector. With the prediction $y_m = \mathrm{argmax}_c \mathbb{P}_m(Y = c \mid X = \boldsymbol{x})$, our second objective is given as

$$\mathcal{L}_2 = \mathbb{E}_{\boldsymbol{x}} \mathbb{E}_{\tilde{z}_{\boldsymbol{x}}} \Big[ \mathrm{CE}\left(y_m, \mathcal{G}(\tilde{\boldsymbol{z}}_{\boldsymbol{x}} \odot \boldsymbol{x})\right) \Big]. \tag{4}$$

**The final objective.** We parametrize $\mathcal{E}, \mathcal{S}$ and $\mathcal{G}$ with three neural networks of appropriate capacity. All networks $\mathcal{E}, \mathcal{S}$ and $\mathcal{G}$ are jointly optimized over total parameters $\theta$ and globally on the training set. We further introduce a regularization term over $\boldsymbol{W}$ to encourage sparsity and accordingly compact explanations. The final objective function is now given as

$$\min_\theta \Big[ \mathcal{L}_1 + \alpha\, \mathcal{L}_2 + \beta\, \mathbb{E}_{\boldsymbol{x}}[||\boldsymbol{W}_{\boldsymbol{x}}||_{2,1}] \Big], \tag{5}$$

where $\|\cdot\|_{2,1}$ is the group norm $2,1$, and $\alpha, \beta$ are balancing coefficients on loss terms. $\alpha$ and $\beta$ are subject to tuning since a highly compressed representation can cause information loss and harm faithfulness. Figure 1 summarizes our framework as follows

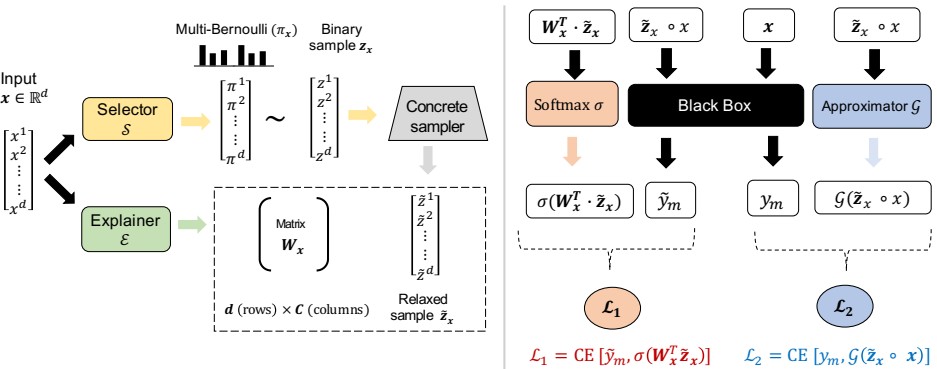

Figure 1: An illustration of AIM pipeline. **Left:** Given an input $\boldsymbol{x}$, the explainer $\mathcal{E}$ produces a local multi-class explanation module $\boldsymbol{W}_{\boldsymbol{x}}$ in which each entry $W_{\boldsymbol{x}}^{i,j}$ representing the relative weight of the $i$th feature of $\boldsymbol{x}$ to the predicted label $j \in \{1, ..., C\}$. $\mathcal{E}$ is optimized on a local space of perturbations around $\boldsymbol{x}$. Such a space is constructed via the feature selector $\mathcal{S}$ that is simultaneously optimized to generate a high-quality local distribution containing well-behaved neighboring samples. The binary sample $\boldsymbol{z}_{\boldsymbol{x}} \sim$ Multi-Bernoulli$(\pi_{\boldsymbol{x}})$ is passed through a Gumbel-Softmax sampler for relaxation. We end up with the explanation matrix $\boldsymbol{W}_{\boldsymbol{x}}$ and relaxed samples $\tilde{\boldsymbol{z}}_{\boldsymbol{x}}$. **Right:** The figure illustrates how these output components interact with each other and the input $\boldsymbol{x}$ to form the first and second loss objectives given in Eq. (2) and (4). The final objective in Eq. (5) combines $\mathcal{L}_1$ and $\mathcal{L}_2$ with an additional sparsity term to induce compactness. CE is the cross-entropy function. $\odot, \cdot,$ and $\sigma$ denote the element-wise product, inner product and softmax operation respectively.

## 3.3 INFERENCE

A standard inference strategy is to choose top $K$ features with the highest weights, with $K$ determined in advance. In our framework, the explainer outputs a weight matrix $\boldsymbol{W}_{\boldsymbol{x}}$ size $d \times C$ (recall that $d$ is the number of features and $C$ is the number of target classes). We obtain the black-box's predicted label $j = y_m = \mathrm{argmax}_c \mathbb{P}_m(Y = c \mid X = \boldsymbol{x})$ and select the corresponding column $W_{\boldsymbol{x}}^{:,j}$ as the weight vector. Features can then be derived accordingly. Though it is intuitive to use $\pi_{\boldsymbol{x}}$ directly for the explanation, doing this may require specifying a certain threshold $\theta \in [0; 1]$. Since $\pi_{\boldsymbol{x}}$ represents

local distributions, choosing the thresholds individually for each input is daunting while setting a global threshold for all inputs is sub-optimal. Moreover, the selection of an $i$th feature using $\pi_{\boldsymbol{x}}$ (i.e., $\pi_{\boldsymbol{x}}^i \geq \theta$) is independent for each feature, so when combined, they do not guarantee the resulting subsets of features can well approximate the black-box's decisions. On the other hand, the explainer looks into input features all-in-once to settle with good subsets of features. Appendix D provides evidence that inference according to $\boldsymbol{W_x}$ is the optimal strategy.

## 4 EXPERIMENTS

We conducted experiments on various machine learning classification tasks. In the main paper, we focus on NLP classifiers since we believe text data is the most challenging modality. In the following, we discuss the experimental design for textual data.

- **Sentiment Analysis:** The Large Movie Review Dataset **IMDB** (Maas et al., 2011) consists of $50,000$ movie reviews with positive and negative sentiments. The black-box classifier is a bidirectional GRU (Chen et al., 2018) that achieves an $85.4\%$ test accuracy.

- **Hate Speech Detection: HateXplain** is an annotated dataset of Twitter and Gab posts for hate speech detection (Mathew et al., 2021). The task is to classify a post either to be normal or to contain hate/offensive speech. The black-box model is a bidirectional LSTM (Gers et al., 2000) stacked under a standard Transformer encoder layer (Vaswani et al., 2017) of $4$ attention heads. The best test accuracy obtained is $69.6\%$.

- **Topic Classification:** AG is a collection of more than 1 million news articles. **AG News** corpus (Zhang et al., 2015) is constructed by selecting $4$ largest classes from the original dataset: World, Sports, Business, and Sci/Tech. We train a word-level convolution neural network (CNN) (LeCun et al., 1995) as a black-box model. It obtains $89.7\%$ accuracy on the test set.

See Appendix A for additional details on our experimental setup and model design. Appendix E further demonstrates the remarkable capability of AIM for generalizing on images and tabular data. Code and data for reproducing our experiments are published at `https://github.com/isVy08/AIM/`.

### 4.1 PERFORMANCE METRICS & BASELINE METHODS

The task of a saliency-based explainer is to find the subset of input features $\mathbb{S}$ that best mimics the black-box's predictions on the original input. Following the suggestions from Robnik-Šikonja & Bohanec (2018) on desiderata of explanations and related works (Ribeiro et al., 2016; Chen et al., 2018; Schwab & Karlen, 2019; Situ et al., 2021; Gat et al., 2022), Table 1 presents the metrics for quantitative evaluation of word-level explanations. See Appendix B for implementation details.

For text classification tasks, we compare our method against baselines that have done extensive experiments on textual data: L2X (Chen et al., 2018), LIME (Ribeiro et al., 2016), VIBI (Bang et al., 2021) and FI (Gat et al., 2022). Regarding model architectures, note that AIM has been intentionally designed to match those of L2X and VIBI to assure fair comparison. For each method, we tune the remaining hyper-parameters over a wide range of settings and report the results for which Faithfulness is highest (See Appendix H).

### 4.2 RESULTS

We compare the performance of methods by assessing the extent to which the set of 10 best features satisfies the criteria discussed in Table 1. Except for AIM and FI that do not treat $K$ as a hyper-parameter, all the other baselines are optimized at $K = 10$. Table 2 reports the average results over 5 model initializations. We here show that our method AIM consistently outperforms the baselines on all metrics while achieving a remarkably high level of faithfulness of over $90\%$ across datasets. AIM effectively approximates the black-box predictions with only 10 features, which demonstrates the sufficiency of the selected feature sets. Given the vast combinatorial space of possible subsets of

---

[2]VIBI (Bang et al., 2021) measures fidelity via the prediction accuracy of the approximator model, whereas we conduct a post-hoc comparison of the black-box's predictions on the original and masked input.

Table 1: Description of quantitative evaluation metrics.

| Property | Definition | Metric | Description |
|---|---|---|---|
| Fidelity | How well does the explanation approximate the prediction of the black box model? | Faithfulness / Post-hoc Accuracy[2] | Degree of agreement between the prediction given the full document and the prediction given the selected words in $\mathbb{S}$. A higher value means the explanations are strongly relevant to the black-box's prediction. |
| Brevity | How concise is the explanation? | Brevity | Number of clusters of duplicates or semantically related words formed over $\mathbb{S}$. A lower value means the tokens are less semantically polarizing and more compact. |
| Comprehensibility | How well do humans understand the explanation? | Purity | Proportion of stopwords and punctuation included in $\mathbb{S}$. A lower proportion is equivalent to a more meaningful feature set. |
| Stability | How similar are the explanation for similar instances? | Intersection over Union (IoU) | Proportion of overlapping words in the explanations of two similar documents. We expect the selected features for two such examples overlap in great quantity. |
| Degree of importance | How well does the explanation reflect the importance of features or parts of the explanation? | Positive $\Delta$ log-odds | Difference in the confidence of the black-box model in a prediction before and after masking important words given in $\mathbb{S}$. A higher value indicates $\mathbb{S}$ contains important features. |
| Degree of importance | How well does the explanation reflect the importance of features or parts of the explanation? | Negative $\Delta$ log-odds | Difference in the confidence of the black-box model in a prediction before and after masking unimportant words i.e., words not in $\mathbb{S}$. A lower value indicates features not contained in $\mathbb{S}$ are unimportant. |

features, we believe the capacity to efficiently search for a sufficient set of features is what makes AIM stand out from the existing works. Examining $\Delta$ log-odds, it is observed that our top 10 features are deemed more important since removing them causes the largest drops in confidence of the black-box model in the original prediction (on the full document). Given an input containing only important features, interestingly there is even a slight increase in confidence when the black-box models make that prediction. Table 2 also reports the average training time (in minutes) for each method. Since AIM is trained in an *instance-wise* manner, AIM matches L2X and VIBI in terms of time efficiency, whereas LIME and FI are extraordinarily time-consuming due to the nature of *additive* models. Learning local explanations instance-wisely also enables AIM to leverage global information, thereby supporting stability (via % IoU) better the baselines.

Table 2: Performance of all methods on 3 datasets at $K = 10$. ↑ Higher is better. ↓ Lower is better.

| Explainer | AIM (ours) | L2X | LIME | VIBI | FI |
|---|---|---|---|---|---|
| | | | IMDB | | |
| **Purity (%)** ↓ | **8.22±0.20** | 12.89±0.27 | 36.55±0.13 | 30.86±0.20 | 30.27±0.86 |
| **Brevity** ↓ | **2.48±0.01** | 2.51±0.14 | 7.73±0.16 | 3.86±0.23 | 3.66±0.75 |
| **Faithfulness (%)** ↑ | **99.62±0.02** | 84.80±0.08 | 79.00±0.21 | 56.80±0.09 | 71.70±0.36 |
| **IoU (%)** ↑ | **6.11±0.09** | 4.50±0.14 | 1.44±0.02 | 0.59±0.01 | 3.04±0.01 |
| **Positive $\Delta$ log-odds** ↑ | **7.53±0.03** | 2.92±0.11 | 2.25±0.18 | 0.09±0.34 | 2.38±2.35 |
| **Negative $\Delta$ log-odds** ↓ | **-0.20±0.05** | 2.63±0.33 | 5.74±0.06 | 8.47±0.36 | 7.15±1.26 |
| **Training time (minutes)** ↓ | 11.19 | **6.90** | 551.42 | 8.48 | 311.42 |
| | | | HateXplain | | |
| **Purity (%)** ↓ | **19.78±2.54** | 21.87±0.14 | 37.73±0.17 | 33.91±0.29 | 33.13±0.09 |
| **Brevity** ↓ | **3.88±0.21** | 4.36±0.15 | 7.59±0.20 | 4.56±0.28 | 4.23±0.00 |
| **Faithfulness (%)** ↑ | **92.98±1.17** | 75.32±0.03 | 80.56±0.17 | 67.25±0.29 | 66.28±0.68 |
| **IoU (%)** ↑ | **6.66±0.30** | 3.42±0.74 | 4.40±0.00 | 2.37±0.08 | 3.04±0.06 |
| **Positive $\Delta$ log-odds** ↑ | **4.98±0.25** | 2.81±0.11 | 3.18±0.11 | 1.41±0.13 | 1.41±0.02 |
| **Negative $\Delta$ log-odds** ↓ | **-1.40±0.16** | 1.15±0.27 | 1.07±0.13 | 2.50±0.11 | 2.59±0.02 |
| **Training time (minutes)** ↓ | 3.13 | 2.43 | 222.17 | **2.15** | 162.17 |
| | | | AG News | | |
| **Purity (%)** ↓ | **3.83±0.05** | 6.64±0.25 | 29.15±0.06 | 21.15±0.25 | 27.61±0.03 |
| **Brevity** ↓ | **3.39±0.00** | 3.94±0.18 | 8.76±0.16 | 4.07±0.03 | 4.91±0.00 |
| **Faithfulness (%)** ↑ | **97.92±0.05** | 90.13±0.26 | 86.64±0.10 | 66.58±0.36 | 76.10±0.11 |
| **IoU (%)** ↑ | **6.48±0.00** | 6.01±0.45 | 3.52±0.02 | 1.68±0.02 | 3.18±0.03 |
| **Positive $\Delta$ log-odds** ↑ | **7.14±0.01** | 4.36±0.28 | 1.38±0.17 | 0.03±0.22 | 0.49±0.02 |
| **Negative $\Delta$ log-odds** ↓ | **-1.09±0.02** | 0.28±0.29 | 0.60±0.13 | 3.88±0.10 | 2.24±0.00 |
| **Training time (minutes)** ↓ | **11.44** | 22.08 | 137.02 | 15.36 | 30.22 |

As $K$ increases, the explanation is expected to be more faithful to the black-box model. Since the mechanism of L2X, VIBI, or LIME requires training a new model with the corresponding $K$, as shown in Figure 2, it may however not guarantee the monotonic behavior. Appendix G analyzes this property in L2X explanations in more detail. We choose to investigate L2X here only since it is the best-performing among the baselines. It is shown that the performance of L2X does not always satisfy monotonicity w.r.t $K$ on a newly trained model: different choices of $K$ can yield different

feature rankings e.g., the features picked by a model trained on top 5 may be considered irrelevant by one trained on top 10. AIM strictly avoids such inconsistency as our framework is not sensitive to $K$.

Table 3 additionally provides 4 examples of the features chosen by AIM in IMDB dataset. This helps shed light on why the black-box model makes a certain prediction, especially the wrong one. A comprehensive qualitative comparison with the baselines on multiple examples can further be found in Appendix C. [3] While explanations from *additive* models (LIME and FI) are contaminated with a larger volume of neutral words, *instance-wise* methods (L2X and VIBI) tend to select more meaningful features. AIM stands out with the strongest compactness by picking up all duplicates and synonyms without compromising predictive performance. Note that LIME suffers from low brevity mainly because its algorithm extracts unique words as explanations. This also means LIME's feature sets tend to be more diverse than the other methods and thus should be more faithful. Our experiment nevertheless shows that this is not the case.

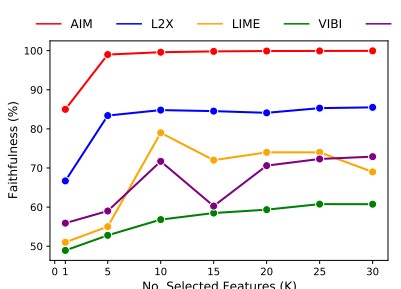

Figure 2: Faithfulness of explanation models at different values of $K$ on IMDB dataset.

Table 3: Ground-truth labels and labels predicted by the black-box model on IMDB movie reviews are given in the first two columns. 10 most relevant words selected by AIM are highlighted in yellow.

| Truth | Model | Key words |
|---|---|---|
| positive | positive | this movie was a pleasant surprise for me. in all honesty, the previews looked horrible, up until the point where emma thompson and alan rickman appeared. so i rented it with reservation, but i thoroughly enjoyed this movie. it had great acting, a few good plot twists, and, of course, emma thompson and alan rickman. it's definitely worth checking out. |
| negative | negative | this may just be the worst movie ever produced. worst plot, worst acting, worst special effects...be prepared if you want to watch this. the only way to get enjoyment out of it is to light a match and burn the tape of it, knowing it will never fall into the hands of any sane person again. |
| positive | negative | to me, "anatomie" is certainly one of the better movies i have seen. i don't think "anatomie" was primarily intended to be a horror movie but a movie questioning the ethics of science. if you watch it with that in mind, it turns into a really good film. the only annoying bit was the awful voice dubbing for the english version. how can you expect any non-german person to listen to these unbearable german accents for two hours ? let native english speakers do the talking or use subtitles instead!! |
| negative | positive | i have seen this movie several times, it sure is one of the cheapest action flicks of the eighties. so, i think many viewers would definitely change the channel when they come across this one. but, if you are into great trash, "dragon hunt" is made for you. the main characters (the mcnamara twins) are sporting great moustaches and look so ridiculous in their camouflage dresses. one of the best scenes is when one of then gets shot in the leg and is still kicking his enemies into nirvana. this movie is really awful, but then again, it is a great party tape! |

## 4.3 HUMAN EVALUATION

We additionally conduct a human experiment to evaluate whether the words selected as an explanation convey sufficient information about the original document to human users. We ask 30 university students to infer the sentiments of 50 IMDB movie reviews, given only 10 key words obtained from an explainer for each review. To avoid confusion, only examples where the black-box model predicts correctly are considered (See Appendix F for the setup). We assess whether the sentiment

Table 4: Human evaluation results on IMDB dataset of AIM, L2X, and LIME.

| Explainer | AIM | L2X | LIME |
|---|---|---|---|
| Human accuracy | 90.10% | 83.03% | 84.13% |
| % Neutral | 8.41% | 12.22% | 19.22% |

inferred by humans is consistent with the actual label of a movie review: *human accuracy*. Some reviews are judged as "neutral / can't decide", because the selected key words are neutral, or because positive and negative words are comparable in quantity. We exclude these neutral examples when computing the average accuracy for a participant, but record the proportion of such examples as a proxy measure for purity. The final accuracy is averaged over multiple participants and reported in

---

[3]All qualitative examples presented in our work are randomly selected from the outputs of the model initialization with the best Faithfulness.

Table 4. It is consistent with our quantitative results that explanations from AIM are perceived to be more informative and contain fewer neural features, thus being more comprehensible to human users.

## 4.4 MULTI-CLASS EXPLANATION

A novel contribution of our work is the capability of simultaneously explaining multiple decision classes from a single matrix $W_x$. Whereas existing methods often require re-training or re-optimization to predict a different class, our explainer produces class-specific explanations in a single forward pass: given a learned $W_x$, select the column $j$ ($W_x^{:,j}$) corresponding to the target class to be explained. To the best of our knowledge, we are the first to propose an explanation module with such a facility.

We assess the quality of multi-class explanations via two modifications of Faithfulness and IoU. The former metric **Class-specific Faithfulness** measures whether the black-box prediction on the explanations aligns with the class being interpreted. The latter **Pairwise IoU** evaluates the overlapping ratio of words in the explanations for a pair of decision classes. Table 5 provides the average results for these metrics, in comparison with LIME and FI. AIM performs surprisingly well on binary classification tasks with the selected feature sets nearly distinctive to each class i.e., overlapping words account only for less than $4\%$. Faithfulness $98.09\%$ of on the first class, for example, means that given the explanations, the black-box model predicts label 0 for $98.09\%$ of testing examples. Meanwhile, the performance of LIME and FI seems to be no better than random and sensitive to the distribution of classes in the datasets. However, the task gets more challenging as more classes are involved. Since AG News is a dataset of news articles from $4$ topics, it is sometimes difficult to clearly distinguish a text between two classes, which we suspect leads to a higher overlapping ratio, thereby harming faithfulness. Regardless, the success on IMDB and HateXplain demonstrates the potential of supporting *counterfactual* explanations that seek to determine which features a black-box classifier attends to when predicting a certain class.

Table 5: Quality of multi-class explanations from AIM, LIME and FI.

| Metric | Class-specific Faithfulness (%) ↑ | | | | Pairwise IoU (%) ↓ |
|---|---|---|---|---|---|
| Target label | 0 | 1 | 2 | 3 | |
| IMDB | | | | | |
| **AIM** | **98.09±0.06** | **98.96±0.05** | - | - | **0.41±0.02** |
| **LIME** | 50.02±0.15 | 50.48±0.16 | - | - | 15.69±0.01 |
| **FI** | 86.32±0.29 | 15.62±0.36 | - | - | 92.34±0.08 |
| HateXplain | | | | | |
| **AIM** | **87.76±0.29** | **88.59±1.88** | - | - | **3.69±0.11** |
| **LIME** | 24.30±0.12 | 74.15±0.11 | - | - | 66.91±0.02 |
| **FI** | 41.21±0.50 | 60.35±0.40 | - | - | 76.52±0.05 |
| AG News | | | | | |
| **AIM** | **73.88±0.47** | **79.13±0.56** | **54.56±0.18** | **84.73±0.38** | **9.24±0.03** |
| **LIME** | 22.89±0.01 | 26.45±0.06 | 25.00±0.02 | 26.71±0.01 | 51.09±0.09 |
| **FI** | 25.07±0.43 | 25.00±0.44 | 26.18±0.20 | 26.71±0.01 | 35.60±0.07 |

## 5 CONCLUSION AND FUTURE WORK

We developed **AIM** - a novel model interpretation framework that integrates local *additivity* with *instance-wise* feature selection. The approach focuses on learning attributions across the target output space, based on which to derive important features maximally faithful to the black-box model being explained. We provide empirical evidence further proving the quality of our explanations: compact yet comprehensive, distinctive to each decision class and comprehensible to human users. Exploring causal or counterfactual explanations, especially within our multi-class module is a potential research avenue. Though extension to regression problems and other modalities such as audio or graphical data is straightforward, our future work will conduct thorough experiments on these modalities along with comprehensive comparisons with related baselines. Furthermore, our paper currently focuses on word-level explanations, so there is a chance of discarding positional or phrasal information (e.g., idioms, phrasal verbs). This can be resolved through chunk-level or sentence-level explanations, which will be tackled in future works of ours.

ACKNOWLEDGMENTS

Trung Le and Dinh Phung were supported by the US Air Force grant FA2386-21-1-4049. Trung Le was also supported by the ECR Seed grant of Faculty of Information Technology, Monash University.

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

## A  EXPERIMENTAL DESIGN

We now discuss the model design for each component in our framework. We parametrize $\mathcal{E}, \mathcal{S}$ and $\mathcal{G}$ by three deep neural network functions. Since our input $X$ is discrete, every network contains a learnable embedding layer. The explainer $\mathcal{E}$ passes the embedded inputs into three 250-dimensional dense layers and outputs $W$ after applying ReLU non-linearity. The selector $\mathcal{S}$ is composed of one bidirectional LSTM of 100 dimension and three dense layers of the same size. Each layer is stacked between a Dropout layer and an activation. The upper layers use ReLU while Sigmoid is a natural choice for the final one. Regarding the network $\mathcal{G}$, after feeding the inputs into its own embedding layer, we process the outputs through a 250-dimensional convolutional layer with kernel size 3, followed by a max-pooling layer over the sequence length. The last layer is a dense layer of dimension 250 together with Softmax activation. We use the same architecture for all tasks and train our model with Adam optimizer at $\tau = 0.2$ and a learning rate of $0.001$. We tune the coefficients $\alpha, \beta$ via grid search to achieve an adequate balance of faithfulness and compression.

Table 6 details data splits and best hyperparameters used in our experiments for 3 text classification (IMDB / HateXplain / AG News) and 2 image recognition tasks (MNIST / Fashion-MNIST). $\alpha$ and $\beta$ are the balancing coefficients on the loss terms in the final training objective. For every dataset, we tune $\alpha$ and $\beta$ via grid search with values in $\{0.1, 0.5, 1, 1.5, 1.8, 2\}$ and $\{1e{-}2, 1e{-}3, 1e{-}4\}$ respectively, and the setting that yields the highest Faithfulness is selected. In Table 12, we provide detailed empirical results showing our superior performance is insensitive to hyper-parameter choices.

Table 6: Dataset statistics and hyperparameters.

| Dataset | Train/Dev/Test | No. of features | $\alpha$ | $\beta$ |
|---|---|---|---|---|
| IMDB | 25000/20000/5000 | 400 | 1.8 | 1e−3 |
| HateXplain | 15000/4119/1029 | 200 | 0.1 | 1e−3 |
| AG News | 120000/6080/1520 | 400 | 0.1 | 1e−4 |
| MNIST | 14000/4623/3147 | 16 | 0.5 | 1e−3 |
| Fashion-MNIST | 15000/3000/3000 | 16 | 0.5 | 1e−3 |

## B  PERFORMANCE METRICS

We here discuss the implementation details of our quantitative metrics for text explanation tasks. Recall that saliency-based approaches produce explanations in the form of a subset of $K$ most important features $\mathbb{S}$ given by the weight vector.

### B.1  PURITY

For text classification tasks, we observe that an explainer sometimes selects stopwords or punctuation as important features, which are incomprehensible from a human user's perspective. An effective explainer should reduce the likelihood of picking such "contaminated" features. Purity quantifies *the proportion of stopwords and punctuation* included in $\mathbb{S}$. We obtain the collection of stopwords and punctuation via NLTK package (Bird, 2006).

### B.2  BREVITY

Given a subset $\mathbb{S}$, we define an explainer achieving brevity if the subset contains closely related features. For textual data, we expect the chosen features to contain a large number of duplicates and/or synonyms. We introduce *cluster ratio* to quantify brevity. Specifically, we first collect a database of semantically related words through WordNet (Miller, 1995). We group tokens in $\mathcal{S}$ into clusters of synonyms (including duplicates), then calculate the average number of clusters formed over $K$ tokens.

### B.3 FAITHFULNESS

Faithfulness measures the degree of agreement between the black-box's prediction given the explanation and the prediction given the original input. When fed into the black-box model, the explanation - a set of discrete features, is reconstructed into a similar representation vector with the original input where features in $\mathbb{S}$ are retained and those not in $\mathbb{S}$ are masked by zero paddings. Faithfulness is a standard criterion to evaluate the quality of textual explanations and commonly adopted in various literature, including our baseline papers (Ribeiro et al., 2016; Chen et al., 2018; Situ et al., 2021; Gat et al., 2022).

### B.4 STABILITY

One desirable property of a good model explainer is Stability - the ability to produce the same explanations given similar examples. In the context of text explanations, the subsets of selected important words are expected to overlap in large quantities for two similar documents. We evaluate explanation stability through a simplified implementation of the measure *Intersection over Union* (**IoU**) originally proposed in (Situ et al., 2021).

Given an example $x$ in the test set, we first search for the nearest neighbors $\mathcal{N}(x)$. The neighboring documents are defined to (1) have the same (black-box predicted) label and (2) be either lexically or semantically similar. We adopt the ratio of overlapping tokens as a proxy metric for lexical similarity. Semantic similarity is measured via cosine similarity of their BERT representations, obtained by summing over the token representations of the last hidden state produced by a pre-trained BERT uncased base open sourced by Hugging Face (Wolf et al., 2019). We then select a set of 20 distinctive neighbors, consisting of top 10 semantically and top 10 lexically similar documents. Let $v_x$ and $v_{x'}$ respectively denote the subsets of top $K$ tokens selected for the instance $x$ and its neighbor $x'$, **IoU** is given as

$$\frac{1}{|\mathcal{N}(x)|} \sum_{x' \in \mathcal{N}(x)} \frac{|v_x \cap v_{x'}|}{|v_x \cup v_{x'}|}.$$

To eliminate the effect of poor initialization, for each model explainer, we evaluate the model initialization with the highest faithfulness and compare the stability of top 10 explanations. Noticing that explainers sometimes favor a large number of stopwords, which may overestimate the measure, we exclude such tokens in the feature sets when computing Stability.

### B.5 $\Delta$ LOG-ODDS

Given an example $x$, $\Delta$ log-odds($x$) measures the change in the confidence of the black-box's prediction before and after masking the features in an explanation (Schwab & Karlen, 2019; Situ et al., 2021). Given the original input vector $\boldsymbol{x}$, the black-box model outputs the predictive distribution $\mathbb{P}_m(Y = c \mid \boldsymbol{x})$ with label $c \in \{1, ..., C\}$. Let $y_m = \text{argmax}_c \mathbb{P}_m(Y = c \mid \boldsymbol{x})$ denote the predicted label.

$$\Delta\text{log-odds} = \text{log-odds}(\mathbb{P}_m(y_m \mid \boldsymbol{x})) - \text{log-odds}(\mathbb{P}_m(y_m \mid \tilde{\boldsymbol{x}})),$$

where log-odds$(\mathbb{P}) = \log \frac{\mathbb{P}}{1-\mathbb{P}}$ and $\tilde{\boldsymbol{x}}$ denotes the masked representation. **Positive** $\Delta$ log-odds refers to the input version where we mask important features i.e., features in $\mathbb{S}$. **Negative** $\Delta$ log-odds refers to the input version where we mask unimportant feature i.e., all features not in $\mathbb{S}$. This is also the input version used to evaluate Faithfulness.

## C  QUALITATIVE COMPARISON

This section presents 12 additional qualitative examples to examine the quality of explanations of all model explainers. These examples are randomly selected from the outputs of the model initialization with the best faithfulness. Examples $9 - 12$ are particularly dedicated to illustrate multi-class explanations. Across all examples, we again demonstrate that our explanations are strongly consistent with black-box's predictions, highly compact (by covering duplicates and synonyms) and distinctive to each decision class.

**1. Original document:** *this movie was a pleasant surprise for me. in all honesty, the previews looked horrible, up until the point where emma thompson and alan rickman appeared. so i rented it with reservation, but i thoroughly enjoyed this movie. it had great acting, a few good plot twists, and, of course, emma thompson and alan rickman. its definitely worth checking out.*

**Truth:** positive - **Model:** positive

| Explainer | Key words |
|---|---|
| AIM | *enjoyed, great, emma, emma, definitely, pleasant, reservation, good, twists, and* |
| L2X | *enjoyed, pleasant, definitely, great, checking, rented, emma, surprise, worth* |
| LIME | *worth, great, enjoyed, and, it, checking, definitely, surprise, movie, thompson* |
| VIBI | *horrible, great, rickman, definitely, honesty, surprise, rented, reservation, the, acting* |
| FI | *horrible, acting, movie, pleasant, great, thoroughly, enjoyed, where, was* |

**2. Original document:** *this may just be the worst movie ever produced. worst plot, worst acting, worst special effects...be prepared if you want to watch this. the only way to get enjoyment out of it is to light a match and burn the tape of it, knowing it will never fall into the hands of any sane person again.*

**Truth:** negative - **Model:** negative

| Explainer | Key words |
|---|---|
| AIM | *worst, worst, worst, worst, plot, acting, any, just, if, to* |
| L2X | *worst, worst, worst, match, any, only, worst, light, plot, tape* |
| LIME | *worst, special, of, acting, match, the, prepared, enjoyment, to, tapeX* |
| VIBI | *burn, worst, knowing, ,, ,, tape, the, the, special, ,* |
| FI | *worst, tape, plot, worst, be, never, burn, worst* |

**3. Original document:** *to me, "anatomie" is certainly one of the better movies i have seen. i dont think "anatomie" was primarily intended to be a horror movie but a movie questioning the ethics of science. if you watch it with that in mind, it turns into a really good film. the only annoying bit was the awful voice dubbing for the english version. how can you expect any non-german person to listen to these unbearable german accents for two hours? let native english speakers do the talking or use subtitles instead!!*

**Truth:** positive - **Model:** negative

| Explainer | Key words |
|---|---|
| AIM | *awful, ?, anatomie, anatomie, unbearable, annoying, dubbing, any, "nt", horror* |
| L2X | *awful, accents, instead, seen, good, do, or, ?, for, !* |
| LIME | *awful, can, non, seen, the, really, speakers, into, instead, annoying* |
| VIBI | *anatomie, questioning, anatomie, german, native, ethics, intended, instead, let, movie* |
| FI | *?, really, me, science, any, better, to, can, let, was* |

**4. Original document:** *i have seen this movie several times, it sure is one of the cheapest action flicks of the eighties. so, i think many viewers would definitely change the channel when they come across this one. but, if you are into great trash, "dragon hunt" is made for you. the main characters (the mcnamara twins) are sporting great moustaches and look so ridiculous in their camouflage dresses. one of the best scenes is when one of then gets shot in the leg and is still kicking his enemies into nirvana. this movie is really awful, but then again, it is a great party tape!*

**Truth:** negative - **Model:** positive

| Explainer | Key words |
|---|---|
| AIM | *great, great, great, nirvana, best, definitely, still, times, and, and* |
| L2X | *seen, great, tape, ridiculous, best, great, great, sporting, awful, !* |
| LIME | *great, is, and, best, party, it, made, the, really, awful* |
| VIBI | *sporting, moustaches, mcnamara, nirvana, into, great, dragon, tape, great, party* |
| FI | *would, great, are, ridiculous, across, have, tape, in* |

**5. Original document:** *this is not a competition against whites we are in this together if you want to criticise other whites this only makes it impossible to ever beat these spics and sand niggers from taking over just as the niggers have .*

**Truth:** hate/offensive speech - **Model:** hate/offensive speech

| Explainer | Key words |
|---|---|
| AIM | *niggers, niggers, sand, whites, whites, a, only, impossible, are, these* |
| L2X | *sand, beat, niggers, spics, niggers, whites, criticise, makes, competition, want* |
| LIME | *niggers, this, whites, are, is, ever, have, makes, we, against* |
| VIBI | *impossible, whites, ever, beat, over, a, niggers, this, to, taking* |
| FI | *if, is, against, this, it, just, spics, niggers, makes* |

**6. Original document:** *all it does is boomerpost while pretending to be a slav and screeches anytime anyone says anything about jews like anyone other than a boomer would be retarded enough to believe slavs do not know their own 20 th century history at the hands of the chosenites .*

**Truth:** normal - **Model:** hate/offensive speech

| Explainer | Key words |
|---|---|
| AIM | *boomer, jews, screeches, retarded, a, a, would, enough, their, hands* |
| L2X | *retarded, jews, th, boomer, 2, enough, slav, pretending, history, screeches* |
| LIME | *jews, believe, all, it, anytime, boomer, hands, boomerpost, retarded, 20* |
| VIBI | *slav, slavs, anyone, own, their, jews, screeches, know, anything, it* |
| FI | *says, be, all, chosenites, 0, a, about, is, pretending* |

**7. Original document:** *peoplesoft chief threatens to sue over oracle statements peoplesofts chief executive accused oracle of spreading misleading information about his stock sales and threatened to sue for defamation .*

**Truth:** business - **Model:** business

| Explainer | Key words |
|---|---|
| AIM | *stock, executive, sales, chief, chief, peoplesoft, peoplesoft, misleading, statements, defamation* |
| L2X | *stock, sales, chief, peoplesoft, executive, "s", statements, peoplesoft, oracle, his* |
| LIME | *his, sales, oracle, chief, executive, about, accused, threatens, misleading, and* |
| VIBI | *defamation, spreading, statements, threatens, executive, sales, oracle, chief, ., peoplesoft* |
| FI | *stock, chief, oracle, accused, peoplesoft, oracle, about, misleading, spreading, chief* |

**8. Original document:** *blunkett gets tougher on drugs new police powers to prosecute offenders for possession if they test positive for drugs when they are arrested, even if the only drugs they have are in their bloodstream, are to be announced this week.*

**Truth:** world - **Model:** sports

| Explainer | Key words |
|---|---|
| AIM | *test, offenders, positive, tougher, when, only, they, they, they, powers* |
| L2X | *blunkett, police, test, possession, only, bloodstream, positive, arrested, offenders, have* |
| LIME | *test, offenders, they, tougher, drugs, the, for, gets, positive, are* |
| VIBI | *prosecute, if, for, are, are, offenders, gets, tougher, the, even* |
| FI | *offenders, they, powers, week, they, drugs, the, test, on, bloodstream* |

**9. Original document:** *fda oks scientist publishing vioxx data (ap) ap - the food and drug administration has given a whistle-blower scientist permission to publish data indicating that as many as 139,000 people had heart attacks that may be linked to vioxx, the scientists lawyer said monday.*

| Explainer | Topic | Key words |
|-----------|-------|-----------|
| AIM | World | *attacks, people, lawyer, permission, linked, heart, ap, ap, whistle, indicating* |
| AIM | Sci/Tech | *scientist, scientist, scientist, data, data, may, many, be, ap, ap* |
| LIME | World | *ap, scientist, data, drug, people, monday, vioxx, attacks, s, may* |
| LIME | Sci/Tech | *monday, vioxx, food, drug, data, said, scientist, people, lawyer, had* |
| FI | World | *may, oks, indicating, administration, vioxx, "s", publishing, -, many, ,* |
| FI | Sci/Tech | *oks, administration, may, drug, "s", ap, scientist, ap, that, many* |

**10. Original document:** *tivo net loss widens; subscribers grow tivo inc.(tivo.o: quote, profile, research) , maker of digital television recorders, on monday said its quarterly net loss widened as it boosted spending to acquire customers, but subscribers to its fee-based tv service rose.*

| Explainer | Topic | Key words |
|-----------|-------|-----------|
| AIM | Business | *profile, quote, rose, quarterly, maker, widened, based, boosted, grow, fee* |
| AIM | Sports | *loss, loss, recorders, widened, television, monday, ;, subscribers, subscribers, but* |
| LIME | Business | *its, quote, digital, profile, widened, loss, said, monday, spending, customers* |
| LIME | Sports | *service, its, research, maker, profile, but, television, rose, monday, boosted* |
| FI | Business | *:, ,, widens, profile, ;, rose, tivo, ), research, tivo.o* |
| FI | Sports | *:, profile, rose, widens, ,, quarterly, it, tivo, digital, its* |

**11. Original document:** *i could not believe the original rating i found when i looked up this film, 9.5? unfortunately it looks like i am not alone. the film, is slow and boring really, one of the sad things is that if the film had been given a realistic rating of around 5 or 6 then the expectation would not have been so high. unfortunately, this was not the case, so when watching the film, and seeing the poor story and acting, i am left giving it a 3/10 score. vinnie jones is superb in lock stock, and also snatch, and he plays a great hard man, however, he should stick to this role. its a bit like when stallone and schwarzenegger have done comedy films, they just don't work. neither can he play lead actor, he plays better as supporting or otherwise. when he plays lead, his acting talents are too 'in view' and shown up as not really very good. mean machine is another good example of this.*

| Explainer | Sentiment | Key words |
|---|---|---|
| AIM | Negative | *poor, unfortunately, boring, 3/10, ?, otherwise, looks, acting, acting, looked* |
| AIM | Positive | *superb, great, also, very, bit, realistic, plays, plays, plays, supporting* |
| LIME | Negative | *poor, work, acting, given, lead, sad, one, rating, comedy, too* |
| LIME | Positive | *the, good, plays, when, not, unfortunately, acting, boring, then, case* |
| FI | Negative | *another, is, unfortunately, not, poor, very, schwarzenegger, otherwise, given, slow* |
| FI | Positive | *another, is, unfortunately, poor, very, not, schwarzenegger, given, otherwise, slow* |

**12. Original document:** *the finest short i've ever seen. some commentators suggest it might have been lengthened, due to the density of insight it offers. there's irony in that comment and little merit. the acting is all up to noonan and he carries his thankless character perfectly. i might have preferred that the narrator be less "recognizable", but the gravitas lent is pitch perfect. this is a short for people who read, for those whose "bar" is set high and for those who recognize that living in a culture that celebrates stupidity and banality can forge contrary and bitter defenders of beauty. a beautiful short film. fwiw: i was pleased at the picasso reference, since i once believed that picasso was just another art whore with little talent; like, i assume, most people - until the day i saw some drawings he made when he was 12. picasso was a finer draftsman and a brilliant artist at that age than many artists will ever become in a lifetime. i understood immediately why he had to make the art he became known for.*

| Explainer | Sentiment | Key words |
|---|---|---|
| AIM | Negative | *stupidity, talent, acting, suggest, just, been, banality, make, might, might* |
| AIM | Positive | *perfect, brilliant, perfectly, artists, beautiful, finest, pleased, celebrates, beauty, reference* |
| LIME | Negative | *and, he, ever, offers, a, it, short, defenders, in, saw* |
| LIME | Positive | *a, and, who, became, beautiful, he, is, short, the, i* |
| FI | Negative | *merit, suggest, immediately, acting, offers, pitch, insight, ., is* |
| FI | Positive | *merit, suggest, immediately, acting, pitch, offers, ., is, insight* |

# D   ABLATION STUDY

Generally, our framework involves both an explainer and a feature selector. The explainer $\mathcal{E}$ aims to produce a multi-class explanation module $\boldsymbol{W_x}$ directly used to infer features. The role of the selector $\mathcal{S}$ is to learn good local distributions to generate high-quality local perturbations to train $\mathcal{E}$. Here we study various setups for AIM to demonstrate that the proposed method yields the optimal performance. We seek to answer the following questions:

1. Does inference from the probability vector $\pi_{\boldsymbol{x}}$ of the selector $\mathcal{S}$ give a better result than inference from $\boldsymbol{W_x}$ of the explainer $\mathcal{E}$?

2. Is the explainer $\mathcal{E}$ a necessary component?

3. Is using samples from learnable local distributions better than using heuristic samples?

To validate these hypotheses, we analyze 3 different approaches on IMDB dataset. Table 7 compares the quality of explanations produced under these setups with the performance level achieved under our proposed method on 3 metrics: Faithfulness, Purity and Brevity.

## D.1   INFERENCE FROM SELECTOR

In the original framework, we experiment with the strategy of inferring features from the output probability vector $\pi_{\boldsymbol{x}}$. Following the same approach, we rank features $\pi_{\boldsymbol{x}}$ in a decreasing order and conventionally select the top 10. We initially argue that the selector operates on each feature independently, thus does not guarantee good performance when combining features into a single explanation. Table 7 supports this argument in that this approach leads to a nearly $20\%$ drop in Faithfulness.

## D.2   TRAINING SELECTOR ONLY

The L2X and VIBI frameworks only contain a feature selector from which to accordingly infer explanations. We investigate whether training the explainer jointly is necessary or if the selector simply does the job. Recall our final objective function

$$\min_{\theta}\Big[\mathcal{L}_1 + \alpha\,\mathcal{L}_2 + \beta\,\mathbb{E}_{\boldsymbol{x}}\Big[||\boldsymbol{W_x}||_{2,1}\Big]\Big],$$

where $\mathcal{L}_1$ is used to train the explainer and optimize $\boldsymbol{W_x}$. Removing the role of explainer, we omit $\mathcal{L}_1$ and the third loss term. We then only train the selector according to $\mathcal{L}_2$ with the support of the approximator $\mathcal{G}$. It can be seen from Table 7 that AIM again under-performs under this setup and the performance does not differ much from the first scenario above. We further note that our method employs the Gumbel trick for Bernoulli sampling, while L2X applies it for Categorical sampling over $K$ features. This explains why despite having the Selector only, L2X can still search for better combinations of features.

## D.3 Training Explainer only

Lastly, we analyze the importance of learning local distributions via the selector compared to using heuristic sampling. In our framework, we expect the selector helps mitigate the risk of ill-conditioned local samples observed in LIME. To validate this hypothesis, we modify our framework to mimic LIME: We first exclude the $\mathcal{S}$ and $\mathcal{G}$. To train $\mathcal{E}$, we uniformly sample local perturbations, denoted as $\hat{z}_{\boldsymbol{x}}$, with the number of non-zero elements also uniformly drawn at random.

$\mathcal{E}$ is now optimized purely on $\mathcal{L}_1$, which is modified as

$$\mathcal{L}_1 = \mathbb{E}_{\boldsymbol{x}}\mathbb{E}_{\hat{\boldsymbol{z}}_{\boldsymbol{x}}}\left[\mathrm{CE}\left(\tilde{y}_m, \mathrm{softmax}(\boldsymbol{W}_{\boldsymbol{x}}^T\hat{\boldsymbol{z}}_{\boldsymbol{x}})\right)\right].$$

Table 7 shows that this heuristic approach does significantly worsen the explainer performance. This proves the effectiveness of optimizing the selector $\mathcal{S}$ together with the explainer $\mathcal{E}$ so that $\mathcal{S}$ can assist $\mathcal{E}$ in learning faithful feature attributions and distinctively across decision classes.

Table 7: Ablation study of AIM on IMDB dataset. * Proposed method.

| Method | Purity (%) ↓ | Brevity ↓ | Faithfulness (%) ↑ |
|---|---|---|---|
| **Inference from Explainer *** | **8.22±0.20** | **2.48±0.01** | **99.62±0.02** |
| Inference from Selector | 22.72±1.56 | 3.15±0.14 | 80.23±0.11 |
| Training Selector only | 26.48±0.16 | 3.09±0.05 | 80.71±0.34 |
| Training Explainer only | 29.77±3.79 | 3.96±0.57 | 61.66±1.90 |

## E Experiments on Images and Tabular data

### E.1 Image

We first describe the experiments for interpreting image recognition machine learning systems. The MNIST and Fashion-MNIST dataset respectively consist of $28 \times 28$ gray-scale images of handwritten digits and article clothing images. We train two simple neural networks on a subset of MNIST digits $0, 1, 2$ and Fashion-MNIST images of T-shirt/Trouser/Pullover. Both networks have the same architecture: $2$ convolution layers kernel size $5$ followed by $2$ dense layers with output Softmax activation. The model on MNIST achieves $97.5\%$ test accuracy while that on Fashion-MNIST gains $95.9\%$.

#### E.1.1 Pixel-based Explanation

In this section, we investigate the potential of AIM framework on visual tasks by comparing AIM with two popular image explanation methods: Integrated Gradients (IG) (Zeiler & Fergus, 2014) and Kernel SHAP (Lundberg & Lee, 2017). To keep it consistent with these baselines, we consider each pixel to be a feature and the goal is to find the optimal local subset of pixels $\mathbb{S}$ for each example $x$ that can approximate the black-box prediction on the full image.

We compare Faithfulness scores of top $K$ selected features with $K \in \{200, 300, 400\}$. We again approximate the explanation with the input variant where the features not selected are masked by zeros. The results are averaged over $5$ model initializations. AIM has been shown to be superior on texts and Table 8 here demonstrates that AIM framework can also work reasonably well on images.

#### E.1.2 Superpixel-based Explanation

We now consider groups of pixels as features. We split each image into $4 \times 4$ patches size $7 \times 7$, resulting in a total of $16$ features. Table 9 reports faithfulness of superpixel-based explanations on MNIST and Fashion-MNIST test sets, averaged over $5$ model initializations. Randomly selected examples of various scenarios are additionally presented for qualitative investigation. We find that the selected features are particularly useful to explain wrong decisions in terms of what spurious signals the black-box model relies on to make predictions e.g., the round shape to predict digit $0$, or the rectangular pattern at the bottom to predict a Trouser instead of a T-shirt.

Table 8: Faithfulness (%) of AIM, IG and SHAP on pixel-based explanation.

| $K$ | 200 | 300 | 400 |
|---|---|---|---|
| MNIST | | | |
| AIM | 96.16 | 96.95 | 97.30 |
| Integrated Gradients | 99.10 | 99.52 | 99.52 |
| Kernel SHAP | 67.33 | 67.33 | 67.33 |
| Fashion-MNIST | | | |
| AIM | 90.66 | 93.13 | 97.60 |
| Integrated Gradients | 92.93 | 94.43 | 94.43 |
| Kernel SHAP | 52.30 | 59.63 | 62.77 |

Table 9: Faithfulness (%) of AIM on superpixel-based explanation.

| $K$ | 5 | 8 | 10 |
|---|---|---|---|
| MNIST | 92.67 | 94.60 | 95.08 |
| Fashion-MNIST | 88.46 | 92.23 | 92.42 |

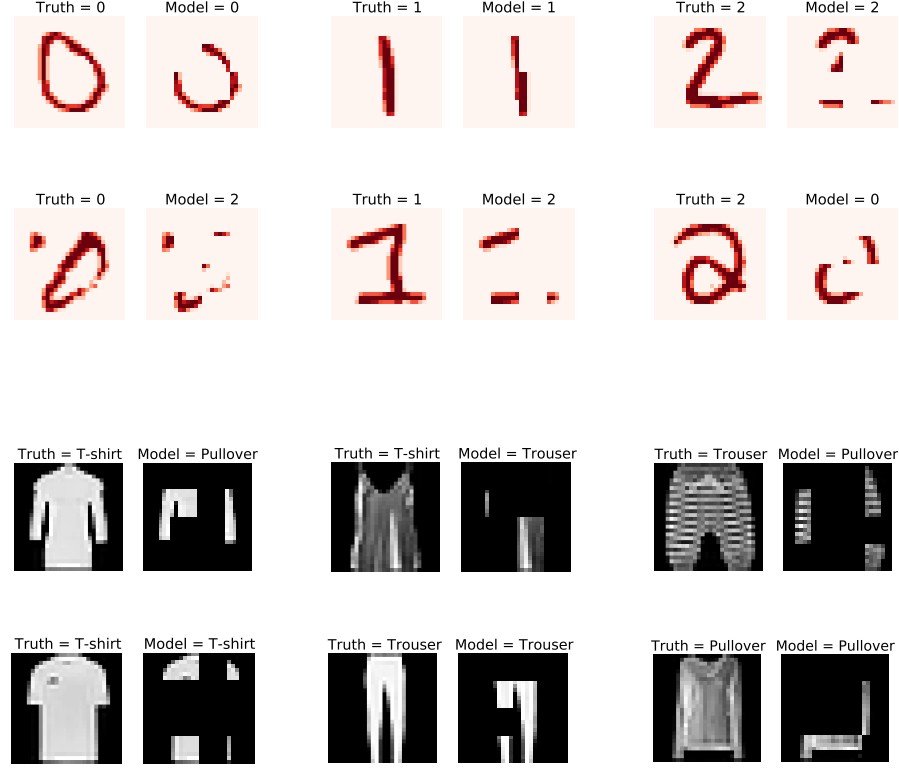

Figure 3: Explanations of the black-box model's predictions based on top 5 most relevant superpixels.

### E.2 TABULAR DATA

We here demonstrate the extensibility of AIM framework on tabular data. We experimented with two real-world datasets: Admission (Acharya et al., 2019) (classifying whether a graduate application is successful on 12 features) and Adult (Kohavi et al., 1996) (predicting whether income exceeds 50K/year on 7 features). For every test example, the task is to select the top most relevant features to the original prediction from a black-box classifier.

**Evaluation metrics.** We want to evaluate how well the top $K$ important features can approximate the prediction on the full input. In the main paper, this is done via the Faithfulness metric: the consistency between the black-box predictions on the full input and the input variant where non-selected features are masked. For texts and images, masking is often done by zero replacement. However, for tabular data, zero values do not indicate the absence of features. We therefore implement *mean masking* and *noise masking* strategies: the former replaces unimportant features with the mean value of each feature over the test set; the latter replaces them with random uniform noise $\epsilon \in [-1, 1]$.

We additionally report the Positive $\Delta$ Log-odds scores, which measure the drop in the black-box confidence scores before and after masking the top $K$ features. A selection of features is deemed important when Faithfulness and Positive $\Delta$ Log-odds metrics are both high.

**Results.** We evaluate the top $K = 6$ and $K = 3$ features (half the number of features) respectively for Admission and Adult datasets. The following table summarizes the average results over 5 model initializations and 10 initializations of random noises. The black-box architectures are given in parentheses. We compare AIM against popular tabular baselines: LIME (Ribeiro et al., 2016), INVASE (Yoon et al., 2019) and MAPLE (Plumb et al., 2018). Here we demonstrate the selected features from AIM do encode sufficient information to yield a prediction consistent with the full input while being more robust to perturbations.

Table 10: Performance of AIM, LIME, MAPLE and INVASE on tabular datasets

| Method | Faithfulness (%) (Mean masking) ↑ | Faithfulness (%) (Noise masking) ↑ | Positive $\Delta$ Log-odds (Mean masking) ↑ | Positive $\Delta$ Log-odds (Noise masking) ↑ |
|---|---|---|---|---|
| Admission ( Random Forest) | | | | |
| AIM | **94.00** | **92.20** | **1.12** | **1.17** |
| LIME | 93.00 | 91.20 | 0.99 | 1.16 |
| MAPLE | **94.00** | 91.30 | 0.74 | 1.03 |
| INVASE | 91.00 | 87.50 | 0.92 | 0.99 |
| Adult (Logistic Regression) | | | | |
| AIM | **99.00** | **86.26** | **1.82** | **1.82** |
| LIME | 86.63 | 83.27 | 1.21 | 1.19 |
| MAPLE | 91.72 | 81.52 | 0.93 | 0.94 |
| INVASE | 81.93 | 67.90 | 0.78 | 0.79 |

## F  HUMAN EVALUATION

We ask 30 university students to infer the sentiments of 50 IMDB movie reviews, each of which is given only 10 key words obtained from each explainer. Each participant is presented with 3 sections, containing output examples from AIM, L2X and LIME respectively. Each section displays 50 sets of 10 key words corresponding to 50 different movie reviews. The information on which section belongs to which method is hidden and the ordering of examples within a section is randomized.

## Movie Reviews Sentiment Evaluation

The survey has 3 sections. In each section, you will be given 50 lines of texts. Each line contains a few words extracted from a movie review. Given only such words, please infer the original sentiment of the movie review ( Positive, Negative, Neutral / Can't decide).

Switch account

* Required

Based on the following words only, guess the original sentiment of the movie review *

| | Positive | Negative | Neutral / Can't decide |
|---|---|---|---|
| 1. [bad, poorly, total, saving, needless, whatever, far, not, not, only] | ○ | ○ | ○ |
| 2. [ridiculous, supposed, plot, plot, no, positive, anything, could, any, any] | ○ | ○ | ○ |
| 3. [awful, bad, bad, bad, horrible, grade, turkey, instead, reason, dialog] | ○ | ○ | ○ |
| 4. [amazing, job, holds, powerful, especially, humanity, message, true, seen, will] | ○ | ○ | ○ |

Figure 4: Human Evaluation Interface

## G    MODEL MONOTONICITY OVER $K$

The following table reports the performance of L2X trained at different values of $K$. It highlights the fact that a careful choice of $K$ as a hyperparameter is crucial, and a larger $K$ does not necessarily yield better results. This is undesirable since in fact, larger $K$ increases the chance of selecting meaningless features (lower purity) while does not guarantee faithfulness will go up accordingly. We also provide qualitative examples showing inconsistencies for selecting top $K$ features i.e, the rankings of features vary across settings. For instance, though qualitatively considered an important feature, the word *amazing* in example 1 is selected in top 5 but does not appear in top 10 and even ranks ninth in top 20. The same pattern can be observed across examples.

Table 11: Performance of L2X when trained at 3 values of $K$ for all datasets. Performance of AIM under the same setup is reported for comparison.

| Explainer | L2X | | | AIM | | |
|---|---|---|---|---|---|---|
| $K$ | Purity (%) ↓ | Brevity ↓ | Faithfulness (%) ↑ | Purity (%) ↓ | Brevity ↓ | Faithfulness (%) ↑ |
| IMDB | | | | | | |
| 5 | 10.99±0.20 | 1.68±0.18 | 83.42±0.08 | 5.91±0.06 | 1.68±0.00 | 99.02±0.03 |
| 10 | 12.89±0.27 | 2.51±0.14 | 84.80±0.08 | 8.22±0.20 | 2.48±0.01 | 99.62±0.02 |
| 20 | 19.23±0.24 | 5.21±0.10 | 84.14±0.06 | 12.46±0.26 | 4.30±0.02 | 99.88±0.01 |
| HateXplain | | | | | | |
| 5 | 22.74±0.16 | 2.65±0.15 | 70.17±0.07 | 13.94±1.84 | 2.59±0.05 | 85.15±0.81 |
| 10 | 21.87±0.14 | 4.36±0.15 | 75.32±0.03 | 19.78±2.54 | 3.88±0.21 | 92.98±1.17 |
| 20 | 26.63±0.10 | 7.25±0.15 | 82.90±0.19 | 18.85±3.70 | 5.38±0.61 | 95.22±1.32 |
| AG News | | | | | | |
| 5 | 4.29±0.18 | 2.39±0.15 | 85.79±0.09 | 1.29±0.03 | 2.10±0.00 | 94.78±0.08 |
| 10 | 6.64±0.25 | 3.94±0.18 | 90.13±0.26 | 3.83±0.05 | 3.39±0.00 | 97.92±0.05 |
| 20 | 13.10±0.20 | 5.99±0.15 | 88.82±0.15 | 13.08±0.03 | 6.53±0.01 | 99.26±0.06 |

**1. Original document:** *i saw Riverdance - the new show and loved it from the very first moment! it is an energetic tribute to Irish dance filled with brilliant dancing, music and choreography! the leads, Jean Butler and Colin Dunne had me captivated with their exquisite dancing! may they always keep shining and keep dancing. their on stage chemistry was amazing, and the unity between them on stage was obvious. they look like they were made to dance with each other! this show is my absolute favourite, and probably always will be. long live Riverdance!*

**Truth:** positive - **Model:** positive

| $K$ | Key words |
|---|---|
| 5 | *loved, favourite, energetic, brilliant, amazing* |
| 10 | *energetic, loved, favourite, tribute, brilliant, first, moment, dancing, butler, obvious* |
| 20 | *energetic, loved, butler, favourite, choreography, !, dancing, moment, amazing, other, dancing, captivated, brilliant, probably, Riverdance, is, ., tribute, music, !* |

**2. Original document:** *how much longer will the west continue to put up with all of this shit from retarded third worlders and liberal cucks before we revert to a frontier mentality and just start the real violence .*

**Truth:** hate/offensive speech - **Model:** hate/offensive speech

| K | Key words |
|---|-----------|
| 5 | *retarded, third, cucks, frontier, violence* |
| 10 | *third, retarded, cucks, worlders, frontier, mentality, violence, liberal, shit, and* |
| 20 | *worlders, third, retarded, and, frontier, mentality, to, from, will, longer, put, a, liberal, the, continue, shit, west, and, much, up* |

**3. Original document:** *Yahoo and SBC extend partnership and plan new services Yahoo and SBC communications have agreed to collaborate to extend some of the online services and content they currently provide to PC users to mobile phones and home entertainment devices.*

**Truth:** sci/tech - **Model:** sci/tech

| K | Key words |
|---|-----------|
| 5 | *Yahoo, Yahoo, users, phones, online* |
| 10 | *phones, online, Yahoo, users, Yahoo, devices, communications, mobile, collaborate, agreed* |
| 20 | *services, entertainment, collaborate, content, extend, and, and, PC, services, Yahoo, extend, some, have, partnership, provide, agreed, home, phones, the, and* |

**4. Original document:** *riding high on the success of "rebel without a cause", came a tidal wave of teen movies. arguably this is one of the best. a very young Mcarthur excels here as the not really too troubled teen. the story concentrates more on perceptions of delinquency, than any traumatic occurrence. the supporting cast is memorable, Frankenheimer directs like an old pro. just a story of a young man that finds others take his actions much too seriously.*

**Truth:** positive - **Model:** positive

| K | Key words |
|---|-----------|
| 5 | *best, memorable, one, troubled, the* |
| 10 | *best, memorable, too, teen, more, ,, riding, high, seriously, young* |
| 20 | *memorable, best, [PAD], very, ., one, high, seriously, troubled, any, perceptions, story, teen, is, young, ., directs, on, traumatic, success* |

**5. Original document:** *i thought i was going to watch another friday the 13th or a halloween rip off, but i was surprised, its about 3 psycho kids who kill, theres not too many movies like that, i can think of mikey, children of the corn and a few others, its not the greatest horror movie but its a least worth a rent.*

**Truth:** negative - **Model:** negative

| K | Key words |
|---|-----------|
| 5 | *surprised, least, 3, greatest, about* |
| 10 | *not, least, surprised, rent, ., i, was, [PAD], thought, worth* |
| 20 | *surprised, least, [PAD], think, halloween, worth, going, ., it, rent, was, mikey, children, a, about, ,, but, "s", [PAD], "s"* |

# H  HYPER-PARAMETER TUNING

## H.1  AIM

In our experiments, the only hyperparameters subject to tuning are loss coefficients $\alpha$ and $\beta$. While $\alpha$ seeks to balance exploration and exploitation of local samples as discussed in the previous sections, $\beta$ controls the magnitude of $||W||_{2,1}$ for stable backpropagation.

We here would like to demonstrate that our framework is not highly sensitive to hyperparameters. As reported, $\beta$ is chosen at $1e-3$ across most text and image datasets, while $\alpha$ can vary within $\{0.1, 0.5, 0.8\}$.

The following table reports the performance of our models under various settings of $\alpha$ (averaged over 5 initializations). It can be seen that there is no significant variation in the performances compared to our reported results (highlighted in bold). For the purpose of clarity, we only display the results for crucial metrics: Faithfulness and $\Delta$ Log-odds scores.

Table 12: AIM Hyper-parameters Tuning.

| $\alpha$ | Faithfulness (%) $\uparrow$ | Positive $\Delta$ log-odds $\uparrow$ | Negative $\Delta$ log-odds $\downarrow$ |
|---|---|---|---|
| | | IMDB | |
| 0.1 | 99.61 | 9.34 | 1.89 |
| 0.5 | 99.56 | 8.83 | -0.09 |
| 1.0 | 99.41 | 6.65 | 0.98 |
| 1.5 | 99.50 | 7.31 | 1.98 |
| **1.8** | **99.62** | **7.53** | **-0.20** |
| 2.0 | 99.10 | 8.35 | 1.49 |
| | | HateXplain | |
| **0.1** | **92.98** | **4.98** | **-1.40** |
| 0.5 | 92.38 | 5.11 | -1.75 |
| 1.0 | 92.50 | 5.87 | -1.82 |
| 1.5 | 92.52 | 5.02 | -1.16 |
| 1.8 | 91.28 | 4.28 | -1.36 |
| 2.0 | 90.67 | 3.16 | -0.44 |
| | | AG News | |
| **0.1** | **97.92** | **7.14** | **-1.09** |
| 0.5 | 96.03 | 7.16 | -1.01 |
| 1.0 | 97.89 | 7.14 | -1.09 |
| 1.5 | 97.03 | 7.10 | -0.88 |
| 1.8 | 97.42 | 7.10 | -1.08 |
| 2.0 | 97.83 | 7.09 | -1.07 |

## H.2  BASELINES

This section provides performance results of the baseline methods under different hyper-parameter settings. Note that our black-box architecture for IMDB dataset is different from ones reported in Chen et al. (2018), Bang et al. (2021) and Gat et al. (2022): L2X adopts CNN while VIBI and FI opt for LSTM. Since AIM is not GRU-based either, our black-box model is chosen to be a bidirectional GRU in order to examine whether these models can explain different kinds of black-box architectures.

The table below lists all of the remaining hyper-parameters subject to tuning and their corresponding model performance. We tune all baselines via grid search over the following ranges and report the average results over 5 initializations. For the purpose of clarity, we only display the results for 3 metrics: Purity, Brevity and Faithfulness. When there is a trade-off among these metrics, Faithfulness is chosen to be the deciding criterion. The best settings for each method are presented in bold and their corresponding results are reported in the main paper.

### H.2.1 L2X

For L2X, we tune $\tau$ which is the Gumbel-Softmax temperature. L2X has only one loss term, so no loss term coefficient needs tuning.

Table 13: L2X Hyper-parameters Tuning.

| $\tau$ | Purity (%) ↓ | Brevity ↓ | Faithfulness (%) ↑ |
|---|---|---|---|
| | IMDB | | |
| 0.1 | 20.15±0.28 | 3.12±0.10 | 83.10±0.35 |
| 0.2 | 11.62±0.18 | 2.46±0.16 | 84.12±0.17 |
| 0.5 | 12.79±0.36 | 2.64±0.21 | 84.40±0.07 |
| **0.7** | **12.89±0.27** | **2.51±0.14** | **84.80±0.08** |
| | HateXplain | | |
| **0.1** | **21.87±0.14** | **4.36±0.15** | **75.32±0.03** |
| 0.2 | 23.52±0.33 | 4.34±0.34 | 75.22±0.33 |
| 0.5 | 17.46±0.15 | 3.48±0.27 | 67.54±0.06 |
| 0.7 | 16.56±0.13 | 2.93±0.06 | 55.78±0.25 |
| | AG News | | |
| 0.1 | 6.56±0.19 | 3.80±0.09 | 89.41±0.08 |
| **0.2** | **6.64±0.25** | **3.94±0.18** | **90.13±0.26** |
| 0.5 | 6.01±0.07 | 3.12±0.10 | 89.08±0.26 |
| 0.7 | 12.54±0.24 | 3.72±0.13 | 87.04±0.12 |

### H.2.2 LIME

The relevant hyper-parameters of LIME for text explanations include kernel width (used to define proximity function) and number of sampling perturbations $N$. As shown in Figure 5, increasing $N$ leads to better faithfulness, yet at the cost of an exponential climb in computing times. At our maximum capacity, we follow the authors' suggestion setting $N = 5000$ for all experiments. We examine kernel width in $\{15, 20, 25, 30, 35\}$.

Table 14: LIME Hyper-parameters Tuning.

| Kernel Width | Purity ↓ | Brevity ↓ | Faithfulness ↑ |
|---|---|---|---|
| | IMDB | | |
| 15 | 35.95±0.07 | 7.43±0.06 | 67.00±0.04 |
| **20** | **36.55±0.13** | **7.73±0.16** | **79.00±0.21** |
| 25 | 36.20±0.20 | 7.34±0.35 | 74.00±0.35 |
| 30 | 37.75±0.15 | 7.39±0.26 | 69.00±0.17 |
| 35 | 37.40±0.37 | 7.57±0.27 | 70.00±0.30 |
| | HateXplain | | |
| 15 | 37.88±0.13 | 7.65±0.11 | 80.27±0.06 |
| 20 | 37.85±0.06 | 7.66±0.37 | 79.49±0.19 |
| 25 | 37.74±0.19 | 7.63±0.14 | 77.94±0.13 |
| 30 | 38.64±0.14 | 7.85±0.06 | 80.27±0.09 |
| **35** | **37.73±0.17** | **7.59±0.20** | **80.56±0.11** |
| | AG News | | |
| 15 | 28.70±0.08 | 8.65±0.09 | 86.12±0.33 |
| 20 | 29.19±0.12 | 8.77±0.11 | 85.46±0.22 |
| **25** | **29.15±0.06** | **8.76±0.16** | **86.64±0.10** |
| 30 | 29.50±0.06 | 8.75±0.14 | 84.87±0.03 |
| 35 | 29.00±0.01 | 8.76±0.02 | 85.46±0.12 |

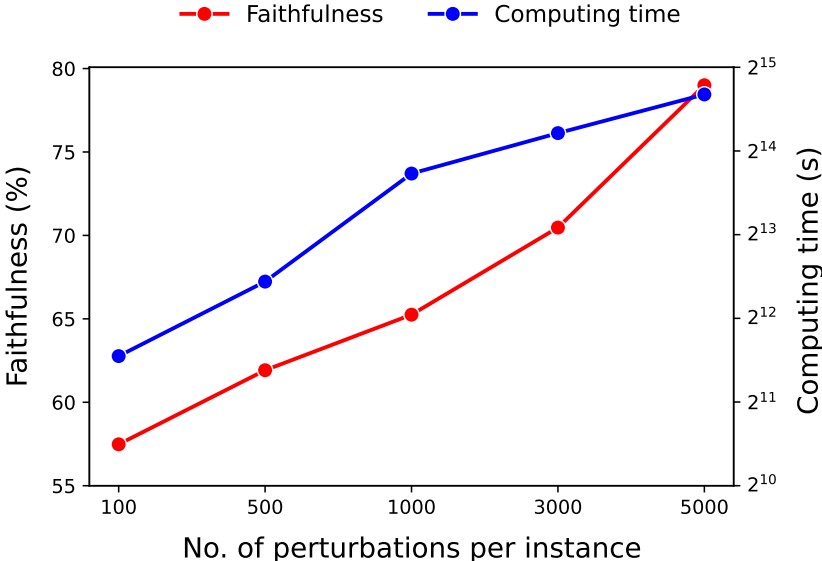

Figure 5: Faithfulness of LIME vs. the number of random perturbations and average processing time on a single CPU over 1000 random test samples in IMDB dataset. The presented model is trained at $K = 10$ and kernel width of 20.

### H.2.3 VIBI

In terms of model architecture, VIBI offers multiple options for the approximator. In our experiments, LSTM approximator gives the highest accuracies for both IMDB and AG News, while CNN works best on HateXplain. For the remaining hyper-parameters, we tune the Gumbel-Softmax temperature $\tau$ within $\{0.2, 0.5, 0.7\}$. The objective function of VIBI further has two loss terms where $\beta$ is the weight of the second one, controlling brevity of the explanation. We explore $\beta$ within $\{0.1, 0.3, 0.5, 1.0, 1.5, 2.0\}$.

### H.2.4 FI

For FI, the relevant hyper-parameter is the number of sampling perturbations $N$. Since FI is very time-expensive, we tune $N$ over 3 values $\{50, 100, 200\}$.

Table 15: VIBI Hyper-parameters Tuning.

| $\tau$ | $\beta$ | Purity (%) $\downarrow$ | Brevity $\downarrow$ | Faithfulness (%) $\uparrow$ |
|---|---|---|---|---|
| | | IMDB | | |
| 0.2 | 0.1 | 30.33±0.20 | 3.81±0.17 | 55.96±0.10 |
| 0.5 | 0.1 | 30.48±0.33 | 3.86±0.14 | 55.64±0.11 |
| 0.7 | 0.1 | 30.20±0.08 | 3.83±0.29 | 56.58±0.35 |
| 0.2 | 0.3 | 30.20±0.34 | 3.82±0.37 | 56.66±0.26 |
| 0.5 | 0.3 | 30.27±0.37 | 3.83±0.35 | 55.80±0.12 |
| 0.7 | 0.3 | 30.59±0.12 | 3.85±0.14 | 55.84±0.10 |
| 0.2 | 0.5 | 30.46±0.06 | 3.82±0.12 | 56.00±0.19 |
| **0.5** | **0.5** | **30.86±0.20** | **3.86±0.23** | **56.80±0.09** |
| 0.7 | 0.5 | 31.10±0.39 | 3.85±0.17 | 55.18±0.04 |
| 0.2 | 1.0 | 30.88±0.33 | 3.90±0.21 | 56.56±0.35 |
| 0.5 | 1.0 | 30.81±0.08 | 3.84±0.15 | 56.34±0.22 |
| 0.7 | 1.0 | 30.90±0.19 | 3.88±0.24 | 55.78±0.15 |
| 0.2 | 1.5 | 30.90±0.26 | 3.87±0.36 | 55.98±0.34 |
| 0.5 | 1.5 | 31.04±0.16 | 3.88±0.41 | 55.82±0.16 |
| 0.7 | 1.5 | 31.21±0.14 | 3.90±0.07 | 55.70±0.36 |
| 0.2 | 2.0 | 31.46±0.23 | 3.89±0.38 | 55.94±0.17 |
| 0.5 | 2.0 | 30.59±0.30 | 3.88±0.21 | 55.56±0.34 |
| 0.7 | 2.0 | 31.12±0.39 | 3.88±0.29 | 55.72±0.13 |

Table 16: VIBI Hyper-parameters Tuning.

| $\tau$ | $\beta$ | Purity (%) $\downarrow$ | Brevity $\downarrow$ | Faithfulness (%) $\uparrow$ |
|---|---|---|---|---|
| | | HateXplain | | |
| 0.2 | 0.1 | 32.77±0.31 | 4.34±0.15 | 66.67±0.14 |
| 0.5 | 0.1 | 33.55±0.15 | 4.37±0.30 | 65.99±0.28 |
| 0.7 | 0.1 | 30.70±0.22 | 4.13±0.40 | 63.95±0.20 |
| 0.2 | 0.3 | 32.77±0.27 | 4.40±0.34 | 65.21±0.06 |
| 0.5 | 0.3 | 24.50±0.15 | 3.44±0.05 | 60.06±0.17 |
| 0.7 | 0.3 | 33.10±0.33 | 4.30±0.13 | 64.43±0.29 |
| 0.2 | 0.5 | 34.63±0.11 | 4.47±0.05 | 65.01±0.07 |
| 0.5 | 0.5 | 30.55±0.33 | 4.10±0.20 | 62.68±0.24 |
| 0.7 | 0.5 | 32.55±0.27 | 4.32±0.32 | 65.99±0.23 |
| 0.2 | 1.0 | 29.69±0.15 | 3.93±0.24 | 64.43±0.12 |
| 0.5 | 1.0 | 33.78±0.25 | 4.48±0.29 | 64.04±0.30 |
| 0.7 | 1.0 | 33.84±0.38 | 4.47±0.31 | 66.57±0.11 |
| 0.2 | 1.5 | 27.39±0.24 | 3.83±0.37 | 61.81±0.38 |
| 0.5 | 1.5 | 35.22±0.26 | 4.52±0.19 | 65.60±0.11 |
| 0.7 | 1.5 | 33.00±0.02 | 4.33±0.18 | 65.99±0.19 |
| 0.2 | 2.0 | 33.45±0.12 | 4.36±0.37 | 66.08±0.10 |
| **0.5** | **2.0** | **33.91±0.29** | **4.56±0.28** | **67.25±0.29** |
| 0.7 | 2.0 | 32.75±0.23 | 4.30±0.28 | 66.47±0.20 |

Table 17: VIBI Hyper-parameters Tuning.

| $\tau$ | $\beta$ | Purity (%) ↓ | Brevity ↓ | Faithfulness (%) ↑ |
|---|---|---|---|---|
| | | AG News | | |
| 0.2 | 0.1 | 19.70±0.05 | 3.99±0.32 | 62.43±0.39 |
| 0.5 | 0.1 | 20.30±0.09 | 4.02±0.29 | 65.66±0.35 |
| **0.7** | **0.1** | **21.15±0.25** | **4.07±0.03** | **66.58±0.36** |
| 0.2 | 0.3 | 19.36±0.12 | 3.82±0.13 | 62.37±0.05 |
| 0.5 | 0.3 | 19.60±0.32 | 3.99±0.06 | 63.55±0.28 |
| 0.7 | 0.3 | 19.68±0.14 | 3.88±0.38 | 63.62±0.02 |
| 0.2 | 0.5 | 20.53±0.23 | 3.95±0.15 | 62.17±0.13 |
| 0.5 | 0.5 | 15.84±0.09 | 3.53±0.20 | 60.26±0.31 |
| 0.7 | 0.5 | 17.66±0.09 | 3.72±0.15 | 61.05±0.24 |
| 0.2 | 1.0 | 21.42±0.32 | 4.08±0.19 | 64.87±0.32 |
| 0.5 | 1.0 | 19.07±0.24 | 3.80±0.03 | 62.76±0.24 |
| 0.7 | 1.0 | 20.41±0.09 | 3.97±0.03 | 61.45±0.25 |
| 0.2 | 1.5 | 18.91±0.32 | 3.87±0.05 | 64.87±0.16 |
| 0.5 | 1.5 | 19.61±0.14 | 3.92±0.25 | 63.75±0.17 |
| 0.7 | 1.5 | 19.44±0.25 | 3.91±0.13 | 61.25±0.18 |
| 0.2 | 2.0 | 20.20±0.25 | 4.00±0.18 | 64.01±0.11 |
| 0.5 | 2.0 | 18.89±0.04 | 3.85±0.04 | 63.88±0.32 |
| 0.7 | 2.0 | 18.14±0.19 | 3.81±0.27 | 62.63±0.03 |

Table 18: FI Hyper-parameters Tuning.

| $N$ | Purity (%) ↓ | Brevity ↓ | Faithfulness (%) ↑ |
|---|---|---|---|
| | IMDB | | |
| 50 | 25.94±0.07 | 3.21±0.01 | 69.26±0.47 |
| **100** | **30.27±0.86** | **3.66±0.75** | **71.70±0.36** |
| 200 | 33.88±0.30 | 3.87±0.28 | 68.78±0.36 |
| | HateXplain | | |
| 50 | 32.84±0.11 | 4.24±0.01 | 65.57±0.81 |
| **100** | **33.13±0.09** | **4.23±0.00** | **66.28±0.68** |
| 200 | 33.09±0.05 | 4.25±0.01 | 66.12±0.31 |
| | AG News | | |
| 50 | 27.62±0.02 | 4.90±0.00 | 75.75±0.08 |
| **100** | **27.61±0.03** | **4.91±0.00** | **76.01±0.11** |
| 200 | 27.63±0.00 | 4.91±0.00 | 76.01±0.06 |

