# OpenReview forum: "An Additive Instance-Wise Approach to Multi-class Model Interpretation"
_ICLR.cc/2023/Conference — ICLR 2023 poster_

### Official Review · Reviewer_Vmy2 · 2022-10-21

**Confidence:** 3
**Correctness:** 3
**Technical Novelty And Significance:** 4
**Empirical Novelty And Significance:** 3
**Recommendation:** 8

**Clarity, Quality, Novelty And Reproducibility:**

The paper is very clearly written and has high quality. The intro and related work contain good overview of the literature and how this paper relates with them. The methodology is mathematically rigorous while being concise at the same time. The definitions of networks and loss functions are well motivated, as they are designed to solve problems in prior works. The experiments are extensive and clear. There are detailed definitions of metrics and many real examples.

The paper is novel in methodology and evaluation metrics.

The paper provides code for their experiment.

**Strength And Weaknesses:**

**Strength**

The paper is clearly written. It is easy to understand and mathematically solid at the same time. The proposed networks and loss functions are very well motivated and make a lot of sense to me. The experiments are extensive: the paper studies several text classification tasks, all  being very useful in practice. There are a number of meaningful evaluation metrics, and the proposed method achieves huge improvement on all of these metrics. In the appendix, there are also extensive studies on loss functions, hyperparameters, and other tasks such as image classification.

**Weakness**

It is unclear to me how $K$ is reflected in training. $K$ is used to define $\mathbb{S}$ but later $x_{\mathbb{S}}$ is approximated by $\tilde{z}_x\odot x$. It is unclear how accurate such approximation is, and it is irrelevant with $K$ anymore. Will constraints like letting $\\|\mathcal{S}(x)\\|_1\approx K$ work?

Some minor things on notation. In eq(2) it would be more rigorous to use $\tilde{y}_m$ instead of $\bar{y}_m$. In addition, the symbol $\circ$ is usually used to represent composition of functions, and the symbol $\odot$ is usually used for element-wise product.

**Summary Of The Paper:**

The paper proposes a new framework for ML interpretation. The goal is to choose a good subset of features (words/pixels) of the input data  to explain its prediction from a black-box classifier. The model contains three networks: a selector that selects important features, an explainer that learns an importance matrix between every feature and class, and an approximator used in the information theoretic lower bound. Through very extensive experiments (mostly on text data), the paper shows improvement of the proposed method compared to baseline methods on multiple evaluation metrics and settings.

**Summary Of The Review:**

Based on the comments above, I think this paper presents an interesting method for feature-based interpretability with extensive evaluation, and therefore I vote for acceptance.

---

> ### Author Response · Authors · 2022-11-14
> **We thank the reviewer for the valuable feedbacks.**
>
> We address the reviewer's questions in the following:
>
> **It is unclear to me how K is reflected in training. K is used to define  S but later $x_S$ is approximated by $z \circ x$. It is unclear how accurate such approximation is, and it is irrelevant with K anymore. Will constraints like $\Vert S(x) \Vert_1 \approx K$ letting  work?**
>
> Let's first recall that given a data example $x$, the Selector learns to produce the probabilities $\pi_x$ where each entry $\pi^i_x$ indicates the probability the $i$th feature is important to the prediction on input $x$.
>
> At one iteration, we sample the binary pattern $z_x \sim \textrm{Multi-Bernoulli}(\pi_x)$ where $z_x^i = 1$ means that the feature $i$ of $x$ is more likely to be relevant to the original prediction (on $x$). $K$ is defined to be the number of important features and specifically here be the number of features with high probability value $\pi_x^i $ (more likely to be sampled as $z_x^i = 1$).
>
> We need to train the network $\mathcal{G}$ on input examples that only contain this subset of $K$ important features. Because input vectors must come in the same size before being fed to a model, we use $z_x \odot x$ to approximate the input examples where only the chosen subset of features are retained their original values while the other features are masked by zero paddings, implying
> the absence of these features. We note that it is a common approximation approach applied across related works, namely [1,2,3,4].
>
> Secondly, unlike many related methods, our framework does not require specifying $K$ nor placing any constraints on choosing $K$ features during training. The number of $K$ important features is flexibly determined by the Selector and varies at different training steps. This means that at one step, the number of important features (those with a probability value near $1$) can be $5$ for example, (equivalent to $K = 5$). However, at later steps, the Selector can converge to more or fewer features, whichever yields a better optimization solution.
>
> [1] Jianbo Chen, Le Song, Martin Wainwright, and Michael Jordan. Learning to explain: An informationtheoretic perspective on model interpretation. In International Conference on Machine Learning,
> pp. 883–892. PMLR, 2018.
>
> [2] Seojin Bang, Pengtao Xie, Heewook Lee, Wei Wu, and Eric Xing. Explaining a black-box by using
> a deep variational information bottleneck approach. In Proceedings of the AAAI Conference on
> Artificial Intelligence, volume 35, pp. 11396–11404, 2021.
>
> [3] Xuelin Situ, Ingrid Zukerman, Cecile Paris, Sameen Maruf, and Gholamreza Haffari. Learning to
> explain: Generating stable explanations fast. In Proceedings of the 59th Annual Meeting of the
> Association for Computational Linguistics and the 11th International Joint Conference on Natural
> Language Processing (Volume 1: Long Papers), pp. 5340–5355, 2021.
>
> [4] Yoon, J., Jordon, J., & van der Schaar, M. (2019, September). INVASE: Instance-wise variable selection using neural networks. In International Conference on Learning Representations.
>
> **In eq(2) it would be more rigorous to use $\tilde{y}_m$ instead of $\bar{y}_m$. In addition, the symbol $\circ$  is usually used to represent composition of functions, and the $\odot$ symbol  is usually used for element-wise product.**
>
> Thank you for helping us improve our work. We have corrected these notations in the revised paper.

---

### Official Review · Reviewer_RWdr · 2022-10-21

**Confidence:** 4
**Correctness:** 3
**Technical Novelty And Significance:** 3
**Empirical Novelty And Significance:** 2
**Recommendation:** 6

**Clarity, Quality, Novelty And Reproducibility:**

- The paper is clearly written. It is easy to understand.
- The proposed method is somewhat novel but hard to say it is super novel because each component of the proposed method is coming from the previous works.

**Details Of Ethics Concerns:**

Not applicable.

**Strength And Weaknesses:**

Strength:
- The proposed method can provide not only instance-wise feature importance, but also corresponding additive explanations. Also it can provide explanation for multiple classes simultaneously.
- The experimental results are promising and consistently better than alternatives.

Weakness:
- The proposed methods can be applicable to any data types. As the authors said, there are many related works on tabular data. In that case, it would be good to provide the superiority of the proposed method on tabular data as well.
- It is unclear how L2 can prevent to converge \pi_x = 1 for all components. It would be great if the authors can explain better on it.
- There are some other related works that also do not need to set K value for instance-wise feature selection (e.g., Yoon et al, 2019). So, it would be good to tone down on this contribution.

**Summary Of The Paper:**

- The authors proposed an interpretable model that combines the additive and instance-wise feature selection approaches.
- More specifically, the proposed methods jointly train three components (explainer, selector, and approximators) to identify important instance-wise features and provide corresponding explanations per each class.
- The proposed methods show consistent performance improvements across multiple text and image datasets over alternatives.

**Summary Of The Review:**

Overall, this paper is easy to read and understand. Proposed method makes sense and the results are promising.
But it would be great if the authors can provide the tabular results to show the superiority over other works in tabular domains.
Also, it would be good to clarify the second loss better.

---

> ### Author Response · Authors · 2022-11-14
> **Experiments on tabular data**
>
> We first thank the reviewer for the time and the valuable comments.
>
> We here demonstrate the extensibility of AIM framework on tabular data. We experimented with two real-world datasets: Admission [1] (classifying whether a graduate application is successful on 12 features) and Adult [2] (predicting whether income exceeds 50K/year on 7 features). For every test example, the task is to select the top most relevant features to the original prediction from a black-box classifier.
>
> **1. Evaluation metrics**
>
> We want to evaluate how well the top $K$ important features can approximate the prediction on the full input. In the main paper, this is done via the **Faithfulness** metric: the consistency between the black-box predictions on the full input and the input variant where non-selected features are masked. For texts and images, masking is often done by zero replacement. However, for tabular data, zero values do not indicate the absence of features. We therefore implement *mean masking* and *noise* masking strategies: the former replaces unimportant features with the mean value of each feature over the test set; the latter replaces them with random uniform noise $\epsilon \in [-1, 1]$.
>
> We additionally report the Positive $\Delta$ Log-odds scores, which measure the drop in the black-box confidence scores before and after masking the top $K$ features. *A selection of features is deemed important when Faithfulness and Positive Log-odds metrics are both high*.
>
> **2. Experimental Results**
>
> We evaluate the top $K = 6$ and $K = 3$ features (half the number of features) respectively for Admission and Adult datasets. The following table summarises the average results over $5$ model initializations and $10$ initializations of random noises. The black-box architectures are given in parentheses. We compare AIM against popular tabular baselines: LIME, INVASE [3] and MAPLE [4], in all of which we use the same architectures and a fixed set of hyper-parameters. Here we demonstrate the selected features from AIM do encode sufficient information to yield a consistent prediction with the full input as well as more robust to perturbations.
>
> | Dataset / Method            | Faithfulness (Mean masking) | Faithfulness (Noise masking) | Positive Log-odds (Mean masking) | Positive Log-odds (Noise masking) |
> |-----------------------------|:---------------------------:|:----------------------------:|:--------------------------------:|:---------------------------------:|
> | Admission ( Random Forest)  |                             |                              |                                  |                                   |
> | AIM                         |          **94.00%**         |          **92.20%**          |             **1.12**             |              **1.17**             |
> | LIME                        |            93.00%           |            91.20%            |               0.99               |                1.16               |
> | MAPLE                       |          **94.00%**         |            91.30%            |               0.74               |                1.03               |
> | INVASE                      |            91.00%           |            87.50%            |               0.92               |                0.99               |
> |                             |                             |                              |                                  |                                   |
> | Adult (Logistic Regression) |                             |                              |                                  |                                   |
> | AIM                         |          **99.00%**         |          **86.26%**          |             **1.82**             |              **1.82**             |
> | LIME                        |            86.63%           |            83.27%            |               1.21               |                1.19               |
> | MAPLE                       |            91.72%           |            81.52%            |               0.93               |                0.94               |
> | INVASE                      |            81.93%           |            67.90%            |               0.78               |                0.79               |
>
> [1] Mohan S Acharya, Asfia Armaan, and Aneeta S Antony. A comparison of regression models for prediction of graduate admissions (2019).
>
> [2] Ron Kohavi, "Scaling Up the Accuracy of Naive-Bayes Classifiers: a Decision-Tree Hybrid" (1996).
>
> [3] Yoon, J., Jordon, J., & van der Schaar, M. INVASE: Instance-wise variable selection using neural networks (2019).
>
> [4] Gregory Plumb, Denali Molitor, and Ameet S Talwalkar. Model agnostic supervised local explana-
> tions (2018).

---

> > ### Author Response · Authors · 2022-11-14
> > **Interaction among AIM components**
> >
> > The following is our response to the question on **how L2 can prevent to converge $\pi_x = 1$ for all components.**
> >
> > Let's first recall that given a data example $x$, the Explainer $\mathcal{E}$ outputs the weight matrix $W_x \in \mathbb{R}^{d \times C}$ with each entry $W^{i,j}_x$ representing the relative weights of feature $i \in \\{1, ..., d\\}$ to the decision class $j \in \\{1, ..., C\\}$. The Selector learns to produce the probabilities $\pi_x$ where each entry $\pi^i_x$ indicates the probability the $i$th feature is important.
> >
> > **We would like to clarify that preventing the probability vector $\pi_x$ from converging to $1$ for all components is the joint effect of all the loss terms (including $\mathcal{L_1}$, $\mathcal{L_2}$, and $ || W_x ||_{2,1}$) from the collaboration of the Selector and the Explainer**. Generally, the Selector tries to offer possibly local good features via maximizing the mutual information w.r.t. the black-box model's original prediction on $x$. We denote this predicted label as $y_m$.
> >
> >
> > Let's first recall our first objective:
> > $$ \mathcal{L_1} =  E_{x} E_{z_{x}} [CE (\bar{y_{m}}, softmax(W_{x}^{T} {z}_{x}) )] $$
> >
> > where CE is the cross-entropy function and $\bar{y_{m}} = argmax \ P_{m} (Y \mid {z}_{x} \odot  x)$.
> >
> >
> > At each iteration, we sample the binary pattern $z_x \sim \textrm{Multi-Bernoulli}(\pi_x)$ where $z_x^i = 1$ means that the feature $i$ of $x$ is offered to the Explainer. Let $\bar{y}_m$ denote the predicted label by the black-box model on $z_x \odot x$.
> >
> > We want the Explainer to predict $z_x$ with the label $\bar{y}_m$ according to the above equation. Since the logit w.r.t. a class $j$ is the sum of all elements in the column $j$ of the matrix $W_x$, to attain the prediction $\bar{y}_m$, the logit of the class $\bar{y}_m$ should be larger than other classes, in other words, the elements in the column $j = \bar{y}_m$ at the selected features (i.e., $z_x^i = 1$) of $W_x$ tend to be higher than other columns.
> >
> > At first, because we initialize the Selector randomly i.e., $\textrm{Multi-Bernoulli}(\pi_x)$ is close to a uniform distribution, the predicted class varies. Gradually, the Selector tends to choose more relevant features to the class $y_m$, and the predicted class $\bar{y_{m}}$ tends to be more stable at $y_m$, making the corresponding logit values stable as well. We note that for the column $j = y_m$, any feature can be offered by the Selector and increased to a higher logit, but the more relevant features and their corresponding elements are more likely to be incremented. As a result, the magnitude of elements in the column $j = y_m$ corresponding to more relevant and important features tends to be higher. Now recall our third term group norm $||W_x||_{2,1}$: minimizing it creates the sparsity every column $j$, hence encouraging the Selector and Explainer to more agree and focus on the more compact set of relevant features, preventing $\pi_x =1$ for all components. We empirically investigate the values in $\pi_x$ and observe that the value of $\pi_x^i$ is high for relevant features while staying low for irrelevant features w.r.t the original prediction.
> >
> > Putting everything together, our AIM framework can output faithful feature sets for individual decision class. Our experimental results in Tables 2 and 5 show that not only can AIM select important features to the original prediction $y$ but it can also identify the suitable and distinctive features to explain the other classes as well. This behavior is also demonstrated by examining qualitative examples 9 and 10 in Appendix C.

---

> > > ### Author Response · Authors · 2022-11-14
> > > **Related Works**
> > >
> > > To the comment : **There are some other related works that also do not need to set K value for instance-wise feature selection (e.g., Yoon et al, 2019). So, it would be good to tone down on this contribution.**
> > >
> > > We are deeply grateful for the updates. We have made the corresponding updates in the Contribution and Related work sections of the revised paper.

---

> > > > ### Comment · Reviewer_RWdr · 2022-12-06
> > > > **Thank you for the detailed response.**
> > > >
> > > > Thank you for the careful answers to my comments.
> > > >
> > > > Overall, my comments are well addressed (especially for the tabular data experiments).
> > > > Please incorporate all the details in this rebuttal to the final camera-ready paper (if accepted).
> > > >
> > > > I would like to raise the score from 6 to 7 after reading the rebuttal.
> > > > However, there is no score 7 that I can select.
> > > > But still this paper is not the level of "clear accept (8)"; thus, I stand on my original score (6).
> > > >
> > > > Thank you!

---

> > > > > ### Author Response · Authors · 2022-12-06
> > > > > **Thank you!**
> > > > >
> > > > > We are deeply grateful for your time and attention to our paper. We have updated the tabular experiments in the revised paper and carefully noted all the suggestions in the second phase of discussion.
> > > > >
> > > > > Thank you!

---

### Official Review · Reviewer_qCbz · 2022-10-22

**Confidence:** 4
**Correctness:** 3
**Technical Novelty And Significance:** 2
**Empirical Novelty And Significance:** 2
**Recommendation:** 3

**Clarity, Quality, Novelty And Reproducibility:**

The method is built on some elements of L2X, but it seems sufficiently novel. My largest concerns are the method design and quality of the writing.

**Strength And Weaknesses:**

--- Strengths ---

- It's an interesting idea to merge selection- and attribution-based methods, as most existing model explanation work falls into one of these categories.
- The AIM method provides fast explanations because it's trained in an amortized fashion (similarly to L2X).
- The experimental results suggest strong performance.

--- Weaknesses ---

Writing and prior work
- The authors frequently refer to two categories of explanation methods, "additive" and "instance-wise." This is poor choice of terminology, neither term seems to mean what the authors intend. "Instance-wise" is synonymous with "local" (i.e., explaining individual predictions), and *all the methods discussed here* are instance-wise/local. And "additive" doesn't cover multiple methods supposedly in that category, such as vanilla gradients (Simonyan et al., 2013) or SmoothGrad (Smilkov et al., 2017). I think the correct terminology would be additive methods --> "attribution methods" and instance-wise --> "selection methods." See [1] for a review of how such methods are related (section 6).
- The authors further conflate instance-wise feature selection with the amortized fashion in which the selectors are often trained (e.g., in L2X). There are multiple parts of the paper revealing this confusion, see the abstract for example: "instance-wise methods directly optimize local feature distributions in a global training framework, thereby being capable of leveraging global information from other inputs." This seems to allude to the explainer training routine, but you don't technically need a trained selector model: you could instead follow a greedy algorithm for each example to be explained, for example.
- The authors mention that prior "instance-wise methods" (read: selection-based methods) have a strict reliance on selecting a fixed number of features. Not true, INVASE and Real-X both address this [2, 3].
- Regarding LIME, the authors write that "there is a chance that the perturbed examples behave undesirably, for example, to change the original prediction." That's not undesirable, it's exactly the point of LIME..... Seeing how the prediction changes helps determine how much the model depends on each feature.
- The authors write that "additive methods are inefficient since they optimize individual explainers for every input." We can't do this for every sentence in the paper, but I'll just pick this one to highlight some more issues. i) Many "additive methods" (read: attribution methods) are not based on optimization, e.g., nothing is being *optimized* on a per-input basis in Integrated Gradients. ii) Some of these methods are inefficient, like LIME, but not all. E.g., most gradient-based methods are on the order of a couple forward passes, so you can't make a blanket statement that they're all slow. iii) Earlier works have addressed the problem of slow attribution methods: [4], [5], and [6] are three examples of works proposing the use of learned explainer models. iv) This suggests that attributions methods are inherently slower than selection methods, but selection methods aren't naturally fast (see comment above about how you don't need a learned selector) - they just happen to have leveraged amortized learning earlier than attribution methods.

[1] Covert et al., "Explaining by Removing: A Unified Framework for Model Explanation" (2021)

[2] Yoon et al, "INVASE: Instance-wise variable selection using neural networks" (2018)

[3] Jethani et al, "Have We Learned to Explain?: How Interpretability Methods Can Learn to Encode Predictions in their Interpretations" (2021)

[4] Dabkowski et al, "Real time image saliency for black box classifiers" (2017)

[5] Schwab & Karlen, "Cxplain: Causal explanations for model interpretation under uncertainty" (2019)

[6] Jethani et al, "FastSHAP: Real-Time Shapley Value Estimation" (2021)

Method
- I can follow the training procedure outlined by the authors, but ultimately I don't understand what we expect the result to be. After implementing the joint training procedure, does the result have anything to do with the information communicated by each feature, for example? Many well-regarded methods (L2X, Integrated Gradients, SHAP) provide some theoretical characterization of their results, including what properties they satisfy or what they represent in terms of information theory. This approach does not, and it has hyperparameters that can significantly affect the results ($\alpha, \beta$) with little thought put into how they should be selected. Explaining ML models is an ambiguous problem with many existing solutions, so at this point solutions with weak theoretical support are unlikely to gain traction.
- It's odd that the method requires training both the selector and explainer, but that the experiments only use the explainer. It seems like this decision was made based on which version of the method performed best in experiments (Appendix D), not by design. It seems a bit wasteful and overcomplicated to train both, and perhaps a flaw in the method that the selector actually hurts the results.
- I suspect that AIM suffers from the same "prediction encoding" problem that [3] exposed in L2X: that the explanation can encode the label via its selections, even with a single feature. If so, this would not be a good thing - it would reflect that the predictions with selected features are based more on which features are selected than the information contained in those features. The results in Table 8 and Table 9 with $K = 5$ show that this may indeed be a problem, because the faithfulness values (% agreement with the full-input prediction) are often >90%. Could the authors comment on this? I think a rigorous test for this issue would involve evaluating the method with $K = 1$, and using the Eval-X method from [3].

Experiments
- Given the abundance of model explanation methods, I don't understand the choice of baselines. L2X I understand, but why not some more common/simple ones like Integrated Gradients [7], occlusion [8] or RISE [9]? SHAP would also be nice to include but has similar computation cost as LIME.
- The authors write that "Except for AIM and FI that do not require specifying $K$, all the other baselines are trained at $K = 10$." LIME doesn't require specifying $K$ either, right? Again, if you wanted methods like L2X that don't require specifying $K$, there are two of them [2, 3].
- Can the authors confirm that their faithfulness and log-odds metrics are calculated using the original classifier rather than the new learned one ($\mathcal{G}$)? For both the text and image datasets?
- The experiments with MNIST and Fashion-MNIST don't compare AIM with any baselines. Can the authors correct this and include some commonly used methods for images (IntGrad, occlusion, RISE)?

[7] Sundararajan et al, "Axiomatic attribution for deep networks" (2017)

[8] Zeiler & Fergus, "Visualizing and understanding convolutional networks" (2014)

[9] Petsiuk et al, "Rise: Randomized input sampling for explanation of black-box models" (2018)

**Summary Of The Paper:**

This work develops a new method to explain predictions from black-box ML models. The method learns two separate modules to perform the following: 1) output an attribution score for each feature and each class (similar to methods like LIME), and 2) output selection probabilities for each feature (similar to methods like L2X). Thus, the method (AIM) is thought to bridge two categories of explanation methods. (The authors refer to these categories as "additive" and "instance-wise", but these are incorrect names for the groups of methods they're referring to.)

The experiments compare the method to L2X, LIME and two other methods (VIBI, FI) and find encouraging results.

**Summary Of The Review:**

The authors propose a new method to combine aspects of existing attribution- and selection-based methods. The method involves a complicated joint training routine with three separate modules, only one of which is used to generate explanations. The method lacks theoretical justification (what the method is optimizing for, e.g., in terms of mutual information) but obtains some positive experimental results.

---

> ### Author Response · Authors · 2022-11-14
> **The different objectives of AIM and Real-X**
>
> **We first confirm our understanding about Real-X and prediction encoding:**
>
> To our best understanding of the paper [3], prediction encoding potentially occurs to amortized explanation methods that involve joint training of a feature selector and a predictor function. Given the selected features, the prediction **obtained from the predictor function of the explainer** agrees strongly with the target label since it can encode the label via the selection, rather than due to the information in the features themselves. In the paper, the task of Real-X is concerned with explaining the ground-truth labels. Since the distribution of the response variable given the selected features is unknown, Eval-X proposes training a surrogate predictor to approximate this distribution. This predictor is later used to evaluate faithfulness of features and investigate prediction encoding.
>
> **We now explain how the goal of AIM is different from Real-X:**
>
> The goal of AIM is to explain the predictions of a black-box classifier to which we assume access, by specifically identifying which features the black-box model relies on to make a certain prediction. This means which features the black-box model exploits may not align from what humans perceive to be linguistically good or relevant. This is one of the many purposes of interpretable machine learning, which is to detect potential biases in machine learning systems.
>
> Therefore, we directly consult this black-box classifier for both training and evaluation. In particular, the evaluation on Faithfulness and Log-odds metrics is conducted with respect to the original black-box classifier (for both text and image datasets). As detailed in Appendix B, during evaluation, each model explainer is only responsible for determining which features to be selected based on its own inference mechanism. For every example, we input the selected features back into the black-box classifier, **not** the learned modules $\mathcal{G}$ or $\mathcal{E}$, to compare the predictive outputs with the full-input prediction (which is too obtained from the black-box classifier).
>
> Our remarkable Faithfulness scores indicate that the selected features from AIM do encode sufficient information to yield a consistent prediction with the full input. We have demonstrated the superior capacity of AIM across various real-world datasets through human assessments and extensive quantitative metrics.

---

> > ### Author Response · Authors · 2022-11-14
> > **Interaction between the Explainer and Selector (1/2)**
> >
> > We here clarify the role of each component and explain what we aim to achieve from joint training.
> >
> > Let's first recall that given a data example $x$, the Explainer $\mathcal{E}$ outputs the weight matrix $W_x \in \mathbb{R}^{d \times C}$ with each entry $W^{i,j}_x$ representing the relative weights of feature $i \in \\{1, ..., d\\}$ to the decision class $j \in \\{1, ..., C\\}$. The Selector learns to produce the probabilities $\pi_x$ where each entry $\pi^i_x$ indicates the probability the $i$th feature is important to the original prediction on $x$.
> >
> > **1. The Selector is by no means a wasteful component since it takes on two important roles:**
> >
> > * (1) the Selector is optimized to generate high-quality local examples approximating local neighbourhood on which the Explainer is trained. We leave the discussion on why learning local distributions is crucial in the section where we explain the drawback of LIME heuristic sampling.
> >
> > * (2) Via the second loss objective, the Selector searches for the optimal subset of features that can approximate the prediction on the full input.
> >
> > **2. How Selector communicates with Explainer:**
> >
> > Let's recall our first objective:
> > $$ \mathcal{L_1} =  E_{x} E_{z_{x}} [CE (\bar{y_{m}}, softmax(W_{x}^{T} {z}_{x}) )] $$
> >
> > where CE is the cross-entropy function and $\bar{y_{m}} = argmax \ P_{m} (Y \mid {z}_{x} \odot  x)$.
> >
> > At each iteration, we sample the binary pattern $z_x \sim \textrm{Multi-Bernoulli}(\pi_x)$ where $z_x^i = 1$ means that the feature $i$ of $x$ is offered to the Explainer. Let $\bar{y}_m$ denote the predicted label by the black-box model on $z_x \odot x$.
> >
> > We want the Explainer to predict $z_x$ with the label $\bar{y}_m$ according to the above equation. Since the logit w.r.t. a class $j$ is the sum of all elements in the column $j$ of the matrix $W_x$, to attain the prediction $\bar{y}_m$, the logit of the class $\bar{y}_m$ should be larger than other classes, in other words, the elements in the column $j = \bar{y}_m$ at the selected features (i.e., $z_x^i = 1$) of $W_x$ tend to be higher than other columns.
> >
> > Let $y_m$ denote the original prediction on $x$. At first, because we initialize the Selector randomly i.e., $\textrm{Multi-Bernoulli}(\pi_x)$ is close to a uniform distribution, the predicted class varies. Gradually, the Selector tends to choose more relevant features to the class $y_m$, and the predicted class $\bar{y_{m}}$ tends to be more stable at $y_m$, making the corresponding logit values stable as well. We note that for the column $j = y_m$, any feature can be offered by the Selector and increased to a higher logit, but the more relevant features and their corresponding elements are more likely to be incremented. As a result, the magnitude of elements in the column $j = y_m$ corresponding to more relevant and important features tends to be higher. Now recall our third term group norm $||W_x||_{2,1}$: minimizing it creates the sparsity every column $j$, hence encouraging the Selector and Explainer to more agree and focus on the more compact set of relevant features.
> >
> > The expected outcome is that we obtain a selection of features encoding sufficient information to predict a certain class. The weight matrix $W_x$ is a compact module containing information about which features are relevant to which classes. Therefore, inference based on $W_x$ is the intended choice, and Tables 2 and 5 demonstrate that our goals have been achieved. Taking multiple decision classes into account, AIM provides an efficient solution to a complex combinatorial search problem.

---

> > > ### Author Response · Authors · 2022-11-14
> > > **Interaction between the Explainer and Selector (2/2)**
> > >
> > > **3. Ablation Study**
> > >
> > > In Appendix D, we further verify our intuition through an extensive ablation study on the IMDB dataset investigating the importance of each component. The method is purely to remove each of them from training and compare changes in the model performance. We here recap the experimental results, which report the Faithfulness and additional Log-odds scores for each variant. *Training Selector only* refers to the variant where we only train the Selector component and use the output probabilities for attributions. *Training Explainer only* refers to the variant we only train the Explainer on uniformly drawn local samples.
> > >
> > > | IMDB                    | Faithfulness (\%) $\uparrow$ | Positive $\Delta$ log-odds $\uparrow$ | Negative $\Delta$ log-odds $\downarrow$ |
> > > |-------------------------|------------------------------|---------------------------------------|-----------------------------------------|
> > > | **Proposed Method**     | **99.62$\pm$0.02**           | **7.53$\pm$0.03**                     | **-0.20$\pm$0.05**                      |
> > > | Training Selector only  | 80.71$\pm$0.34               | 1.96$\pm$0.03                         | 3.55$\pm$0.13                           |
> > > | Training Explainer only | 61.66$\pm$1.90               | 0.21$\pm$0.06                         | 7.69$\pm$0.42
> > >
> > > * Without the Selector, the Explainer only focuses on optimizing the feature coefficients of the linear classifier to predict the correct label for a local example $z_x \circ x$, which do not encode useful information to help top selected features approximate the full-input predictions. These are two different goals, both of which we want to achieve through the first and second loss objective respectively.
> > >
> > > * While the Selector of L2X directly optimizes the Categorical probability of a feature being in the top K important features, our Selector function operates on each input independently through Bernoulli sampling and is not optimized for relative attributions, which is the task of the Explainer. As discussed, one may think of specifying a certain threshold $\theta \in [0;1]$ to do inference. Since $\pi_x$ represents local distributions, choosing the thresholds individually for each input is daunting while setting a global threshold for all inputs is sub-optimal.
> > >
> > > In conclusion, jointly training allows each component to focus on optimizing its own tasks most effectively and our experimental results support our intuition. We have carefully checked our implementation and published our codes for reproduction.

---

> > > > ### Author Response · Authors · 2022-11-14
> > > > **About LIME and Importance of learning local distributions**
> > > >
> > > > Here we explain why one needs a learnable module to optimize for local distributions to generate well-behaved local examples.
> > > >
> > > > First, it is well-reported that LIME suffers from high variance due to perturbation sampling randomness. As the number of features $d$ gets large, realizing the space of $2^d$ possible binary patterns becomes challenging. This set of binary patterns include many patterns involving no or few meaningful features (i.e., features with the pattern 1). It is thus very difficult for such a simple linear separator as one used in LIME to learn useful patterns within finite sampling rounds. LIME mitigates this by adding a distance penalizing term that helps focus on searching examples close to the input (i.e., $\exp(-D(x,z)^2 / \sigma^2)$ in the LIME paper). This term however relies on a careful choice of kernel width $\sigma$.
> > > >
> > > > Second, recall that the role of the Selector $\mathcal{S}$ is two-fold in our framework: (1) it is used to generate local examples approximating local neighbourhood to train the Explainer $\mathcal{E}$; (2) it seeks to search for the optimal subset of features that can locally approximate the full input prediction.
> > > >
> > > > Elaborating on the previous explanations, AIM employs the Selector to balance between exploitation and exploration. The stochastic sampling process yields various examples that produce different predictions, and the Explainer also exploits such variation to learn feature attributions with respect to other classes. At the same time, by optimizing the second objective, we expect the Selector to sample more of the patterns $z_x$ corresponding to the subset of features that can approximate the full-input prediction. This encourages the Explainer to focus more on our main task - finding the optimal combinations of features that explain the original prediction. In this respect, it is thus very difficult for LIME to effectively control for the quality of local samples that would guide the linear model to focus on the features truly explain the original prediction.
> > > >
> > > >
> > > > To your question about **whether LIME requires specifying $K$**: as we examine the published codes of LIME, its explanation function does require an additional argument *num\_features* controlling the number of features present in an explanation. To our understanding, LIME implementation consists of 2 steps:
> > > >
> > > > * Step 1: Identify the top $K$ important features. Some methods include forward selection or training a linear model and selecting the top $K$ with the highest coefficients.
> > > >
> > > > * Step 2: Fit a linear model to the selected top $K$ features (from step 1) and return the final coefficients as attribution scores.

---

> > > > > ### Author Response · Authors · 2022-11-14
> > > > > **Hyper-parameter Tuning and Additive vs. Instance-wise terminologies**
> > > > >
> > > > > **1. Sensitivity of Hyper-parameter Tuning:**
> > > > >
> > > > > In our experiments, the only hyperparameters subject to tuning are loss coefficients $\alpha$ and $\beta$. While $\alpha$ seeks to balance exploration and exploitation of local samples as discussed in the previous sections, $\beta$ controls the magnitude of $||W||_{2,1}$ for stable back-propagation. Tuning these values is a common practice in multi-objective machine learning problems.
> > > > >
> > > > > We would like to demonstrate that our framework is not highly sensitive to hyperparameters. As reported, $\beta$ is chosen at $1e-3$ across most text and image datasets, while $\alpha$ can vary within $\\{0.1, 0.5, 0.8\\}$.
> > > > >
> > > > > The following table reports the performance of our models under various settings of $\alpha$ (averaged over $5$ initializations). It can be seen that there is no significant variation in the performances compared to our reported results (highlighted in bold).
> > > > >
> > > > >
> > > > > |    $\alpha$    | Faithfulness $\uparrow$ | Positive log-odds $\uparrow$ | Negative log-odds $\downarrow$ |
> > > > > |:--------------:|:-----------------------:|:----------------------------:|:------------------------------:|
> > > > > | **IMDB**       |                         |                              |                                |
> > > > > |            0.1 |          99.61%         |             9.34             |              1.89              |
> > > > > |            0.5 |          99.56%         |             8.83             |              -0.09             |
> > > > > |            1.0 |          99.41%         |             6.65             |              0.98              |
> > > > > |            1.5 |          99.50%         |             7.31             |              1.98              |
> > > > > |        **1.8** |        **99.62%**       |           **7.53**           |            **-0.20**           |
> > > > > |            2.0 |          99.10%         |             8.35             |              1.49              |
> > > > > | **HateXplain** |                         |                              |                                |
> > > > > |        **0.1** |        **92.98%**       |           **4.98**           |            **-1.40**           |
> > > > > |            0.5 |          92.38%         |             5.11             |              -1.75             |
> > > > > |            1.0 |          92.50%         |             5.87             |              -1.82             |
> > > > > |            1.5 |          92.52%         |             5.02             |              -1.16             |
> > > > > |            1.8 |          91.28%         |             4.28             |              -1.36             |
> > > > > |            2.0 |          90.67%         |             3.16             |              -0.44             |
> > > > > | **AG News**    |                         |                              |                                |
> > > > > |        **0.1** |        **97.92%**       |           **7.14**           |            **-1.09**           |
> > > > > |            0.5 |          96.03%         |             7.16             |              -1.01             |
> > > > > |            1.0 |          97.89%         |             7.14             |              -1.09             |
> > > > > |            1.5 |          97.03%         |             7.10             |              -0.88             |
> > > > > |            1.8 |          97.42%         |             7.10             |              -1.08             |
> > > > > |            2.0 |          97.83%         |             7.09             |              -1.07             |
> > > > >
> > > > >
> > > > > **2. Clarification of Additive and Instance-wise terminologies:**
> > > > >
> > > > > Our adoption of these terms follows from L2X paper (Chen et al., 2018). Under the term "Instance-wise", we refer to methods adopting amortized or global optimization framework (e.g., L2X) that can exploit information from other examples when explaining an input example. Using the term "Additive", we refer to methods that produce explanations for each individual instance example separately (e.g., LIME). In the respect of efficiency, we compare time efficiency when producing post-hoc explanations from "instance-wise" and "instance-specific" explanation fashions: "instance-wise" methods can explain multiple examples on the spot post-training, while most "instance-specific" methods involve per-input computation / processing to yield an explanation.
> > > > >
> > > > > However, since many "instance-wise" methods happen to adopt the "selection" technique while "additive" methods mostly focus on "attribution", we entirely agree that this is a confusing way to categorize works that confuses many aspects with one another. We further acknowledge the overstatement regarding the reliance on number of features of methods and deeply thank the reviewer for the updates on related works. The related sections have been updated to clarify the confusion and distinguish methods more thoroughly.

---

> > > > > > ### Author Response · Authors · 2022-11-14
> > > > > > **Additional baselines**
> > > > > >
> > > > > > In the main paper, we focus on model interpretability for textual data. Therefore, we only compare AIM with baseline that have done extensive experiments on texts, while Integrated Gradients, RISE or SHAP are originally proposed for images. In Appendix E, we provide experimental results on image recognition tasks, mainly to demonstrate extensibility of AIM framework.
> > > > > >
> > > > > > We have conducted additional experiments investigating the potential of AIM for images and that of Integrated Gradients (IG) and SHAP for text explanation. While IG and SHAP can explain images effectively, they are not ideal for textual tasks. We believe the discrete nature of text features makes it a more challenging modality. AIM has been shown to be superior on texts and we here demonstrate that AIM framework can also work reasonable well on images.
> > > > > >
> > > > > > Details on the experiments are given in the following. The results are averaged over $5$ model initializations.
> > > > > >
> > > > > > **1. Image Explanation**
> > > > > >
> > > > > > For images, the task is to explain a CNN model classifying digits 0,1,2 of the MNIST dataset. We reuse the setup of black-box architecture and hyper-parameters described in Appendix E. To keep it consistent with these baselines, we now consider each pixel to be a feature and the goal is to find the optimal local subset of pixels $\mathbb{S}$ for each example $x$ that can approximate the black-box prediction on the full image.
> > > > > >
> > > > > > We compare Faithfulness scores of top $K$ selected features with $K \in \\{200, 300, 400\\}$. Note again that Faithfulness is evaluated with respect to the black-box classifier, and we again approximate $\mathbb{S}$ with the input variant $x_\mathbb{S}$ where features not in $\mathbb{S}$ are masked by zeros (black pixels).
> > > > > >
> > > > > > | Method                | K = 200 | K = 300 | K = 400 |
> > > > > > |-----------------------|:-------:|:-------:|:-------:|
> > > > > > | AIM                   |  96.16% |  96.95% |  97.30% |
> > > > > > | Integrated Gradients |  **99.10%** | **99.52%** | **99.52%** |
> > > > > > | SHAP                  |  67.33% |  67.53% |  68.20% |
> > > > > >
> > > > > > **2. Text Explanation**
> > > > > >
> > > > > > We adopt Omnixai libray for reliable adaption of IG on texts. Due to the limit of time and complexity in adapting SHAP's codes to non-Transformers model, we only evaluate SHAP on IMDB dataset. We analyze the quality of top 10 selected features of each method using the 3 important metrics: Faithfulness, Positive $\Delta$ Log-odds and Negative $\Delta$ Log-odds. We recap results on AIM for comparison.
> > > > > >
> > > > > > | Method               | Faithfulness $\uparrow$ | Positive log-odds $\uparrow$ | Negative log-odds $\downarrow$ |
> > > > > > |----------------------|:-----------------------:|:----------------------------:|:------------------------------:|
> > > > > > |         **IMDB**         |                         |                              |                                |
> > > > > > | AIM                  |        **99.62%**       |           **7.53**           |            **-0.20**           |
> > > > > > | Integrated Gradients |          55.96%         |             0.11             |              8.47              |
> > > > > > | SHAP                 |          51.76%         |             0.89             |              1.21              |
> > > > > > |      **HateXplain**      |                         |                              |                                |
> > > > > > | AIM                  |        **92.98%**       |           **4.98**           |            **-1.40**           |
> > > > > > | Integrated Gradients |          60.74%         |            1.25            |             3.30            |
> > > > > > |        **AG News**       |                         |                              |                                |
> > > > > > | AIM                  |        **97.92%**       |           **7.14**           |            **-1.09**           |
> > > > > > | Integrated Gradients |          65.59%         |            0.09            |             3.89             |

---

### Author Response · Authors · 2022-12-05
**Review the theoretical perspective of LIME, SHAP, L2X (1/2)**

### We thank the reviewers for the very insightful discussions. Regarding the recent questions from Reviewer qCbz, we believe it is a critical discussion. Therefore, for the sake of clarity, we would like to post our responses here.

Let's first recall our setup: given a data example $x$, the Explainer $\mathcal{E}$ outputs the weight matrix $W_x \in \mathbb{R}^{d \times C}$ with each entry $W^{i,j}_x$ representing the relative weights of feature $i \in \\{1, ..., d\\}$ to the decision class $j \in \\{1, ..., C\\}$. The Selector $\mathcal{S}$ learns to produce the probabilities $\pi_x$ where each entry $\pi^i_x$ indicates the probability the $i$th feature is important to explain the original prediction. We employ the selector $\mathcal{S}$ to sample the binary vector $z_x \sim$  MultiBernoulli$(\pi_x)$ to define the local neighbourhood around $x$. Let $f$ denote the black-box classifier under analysis where $f(x) = \log \mathbb{P}_m(Y = c \mid x)$ with $c \in \\{1, ..., C\\}$. Let

To keep the notations consistent throughout, we denote

* $z \in \\{0,1\\}^d$ as a binary vector for the interpretable (simplified) representation of $x$ indicating the absence/presence of a feature.

* $h_x(z)$ as a mapping of a binary vector into the original input space $\mathbb{R}^d$. $x' = h_x(z)$ refers to the simplified input where only a subset of features have their input values retained.

**Q1. The main one is that we don't know what this method does from a theoretical perspective. This is in contrast with existing methods (L2X, LIME, SHAP), and it makes it impossible for users to reason about the scores.**

According to the classification in the SHAP paper, similar to LIME, our AIM can be categorized into Additive Feature Attribution methods where a linear model is used to approximate the black-box model in a local neighbourhood of $x$.
More specifically, by examining the term $W_{x}^{T} z_x$ in the first objective and setting $\phi$ as the $c$-th column vector in the weight matrix $W_x$ ($c$ is a predicted class), we have a linear model  $g(z_x) = \phi^0_{x} + \sum_{i=1}^d \phi^{i}_{x} z^i_x$, aiming to locally approximate $f$ at prediction $c$.

In the paper, LIME provides a rigorous discussion about desirable properties of a good explainer including local fidelity and model-agnosticism. Apart from the strong and convincing motivations for developing LIME, it appears that there was not any theoretical guarantee. Subsequently, the SHAP paper mentioned three properties uniquely determining additive feature attributions which are Property 1 (Local accuracy) (i.e., the explanation model matches the prediction of $f$ on simplified input $x'$ of $x$: $g(z_x) = f(x')$), Property 2 (Missingness) (i.e., missing features have no impact), and Property 3 (Consistency) (i.e., the consistency of the explanations for two black-box models $f,f'$).

LIME's objective function includes a weighting kernel  $\pi_{x}\left(x'\right)=e^{-D(x,x')^{2}/\sigma^{2}}$ implicitly enforcing higher weights to closer samples. To avoid confusion with our selection probability, we denote the weighting kernel $\pi_x$ to be $\lambda_{x}$.

It was concluded in the SHAP paper that methods not based on Shapley values violate local accuracy and/or consistency (e.g., the original version of LIME). Moreover, also in the SHAP paper, as shown in Theorem 2,  a modified LIME with regularization $\Omega(g) =0 $ and the weighting kernel $\lambda_{x}(x')= \frac{d-1}{C^{|x'|}_d|x'|(d-|x'|)}$ can recover the the Shapley values. However, this is different from the setting of LIME and not used in practice.

Moreover, the Shapley values used in the SHAP paper is originated from the cooperative game theory. The value $\phi^i$ used to measure the difference when the party $i$ is involved or not involved is defined as follows:

$$
\phi^{i}=\sum_{\mathbb{S} \subseteq \mathbb{F} \backslash \{i\}}\frac{|\mathbb{S}|!(|\mathbb{F}|-|\mathbb{S}|-1)!}{|\mathbb{F}|!}\left[f\left(x_{\mathbb{S}\cup\{i\}}\right)-f\left(x_{\mathbb{S}}\right)\right]
$$

---

> ### Author Response · Authors · 2022-12-05
> **Review the theoretical perspective of LIME, SHAP, L2X (2/2)**
>
> In the context of black-box model explanation, $f$ is a black-box model and $\mathbb{F}$ is the set of all features. $\mathbb{S} \subseteq \mathbb{F}$ thus denotes feature subsets and $x_{\mathbb{S}}$ represents the values of the input features in the set. Therefore, $\phi^i$ measures the difference in the model outputs when we include or exclude the attribution or feature $i$. This formulation totally makes sense in the context of cooperative game theory where we need to justify the involvement and importance of the party $i$ itself alone. Although the adoption of $\phi^i$ in the context of model explanation is reasonable, it seems to have a limitation in some cases. Concretely, for text data, the importance and influence of a word are contextually dependent on other words, meaning that a specific word when combining with other relevant words might be more important. The purpose of text explainers is to select a bag of words that well explains and characterises the predictions of text classifiers.
>
> It might be arguable that the above formulation considers all possible combinations of an arbitrary subset $\mathbb{S}$ and $i$, hence also containing meaningful combinations with $i$. However, the number of meaningful combinations are significantly smaller than arbitrary combinations. Therefore, it stands on reason to say that $\phi^i$ does not focus and characterize meaningful combinations.
>
>
> Moreover, it is computationally infeasible to compute $\phi^i$ accurately. Hence, Shapley sampling or approximation approaches need to be resorted for estimating the Shapley values. In the SHAP paper, it seems that there was not any analysis about the variance of sampling or tightness of approximation. Particularly, the sampling approach has very high variance for large $d$ due to $2^{d-1}$ subsets, hence possibly resulting different scores each time. According to Theorem 2 about Shapley kernel, we can solve a linear regression problem using a closed-form to analytically work out Shapley values. However, the complexity of this approach is $O(2^d + d^3)$ which is extremely inefficient for sufficient large dimension $d$. Another workaround is to use Deep SHAP wherein DeepLift with the reference values replaced by the input averages is used. As mentioned in the SHAP paper, *DeepLIFT approximates SHAP values assuming that
> the input features are independent of one another and the deep model is linear*. However, it is unclear about the tightness of the estimation gap for non-linear deep nets. Certainly, if Deep SHAP cannot approximate precisely the Shapley values, Property 3 (Consistency) is violated.
>
> Regarding L2X, Theorem 1 is a classical result in variational inference. However, solving (1) or (2) in that paper directly is impossible. We need to maximize a lower bound instead. Particularly, the quality of the solution depends on the complexity of the deep net used to define $Q_{\mathbb{S}}(Y \mid x_{\mathbb{S}})$. The stochastic optimization process certainly introduces more variance to the training process and final solution. We observe that when training L2X explainers, we need to indicate the hyper-parameter $K$. In Appendix G, we study the qualitative and quantitative results of L2X with different $K$, which shows that the outputs of L2X are not consistent across different $K$.

---

> > ### Author Response · Authors · 2022-12-05
> > **Theoretical perspective of AIM**
> >
> > We here provide the theoretical perspective of AIM. Given an example $x$ and a predicted class $c$, we consider the column $c$ of the weight matrix $W_x$ and denote it by $\phi(x) = [\phi^1(x),...,\phi^d(x) ]^T$, where $\phi(x)$ is a learnable deep net. We train a linear model (i.e., linear according to its weights) w.r.t the prediction $c$ as
> >
> > $$ g_{c} (z_x) = \phi^0(x) + \sum_{i=1}^d\phi^i(x)z^{i}_{x} $$
> >
> > to approximate $f_c(h_x(z_x))$ where $z_x \in \{0,1\}^d$ and $h_x(z_x) = z_x \odot x$. Let $f_c(z_x) = f_c(h_x(z_x))$. If $\phi(x)$ is sufficiently expressive, we could gain a good approximation $g_c(z_x)$ for $f_c(z)$ (i.e., $g(z_x) \approx f_c(z_x)$).
> >
> > We first explain how AIM can approximately satisfy the properties proposed in the SHAP paper.
> >
> > *i) Property 1: Local Accuracy*
> > $$g_c(z_x) \approx f_c(h_x(z_x)) = f_c(z_x)$$
> >
> > This property says that the explanation model $g(z_x)$ matches the prediction on simplified input $x' = z_x \odot x$. AIM meets this property by definition and directly through our first training objective.
> >
> > *ii) Property 2: Missingness*
> > $$z^i_x = 0 \Longrightarrow \phi^i = 0$$
> >
> > Missingness says that a missing or absent feature has no attribution. Citing from an author of SHAP paper, Molnar (2018) pointed out that this is not strictly applied. In theory, an absent feature can have an arbitrary attribution score without hurting the local accuracy property because it is multiplied with $z^i_x = 0$. AIM therefore obeys this property.
> >
> > *iii) Property 3: Consistency*
> >
> > Given an input $z \in \{0,1\}^d$, let $f_x(z) = f_c(z) = f_c(h_x(z))$ and $z \backslash i$ denote the setting where $z^{i} = 0$. We have
> > $$
> > f_x(z) - f_x(z \backslash i) \approx g_c(z) - g_c(z \backslash i) = \phi^i(x) z^i.
> > $$
> >
> > Similarly, if we have another black-box model $f'(z)$ and linear model $g'(z)$ then
> >
> > $$
> > f'_x(z) - f'_x(z \backslash i) \approx g'(z) - g'(z \backslash i) = \phi'^i(x)z^i.
> > $$
> >
> > Finally, if $f'_x(z) - f'_x(z \backslash i) \geq f_x(z) - f_x(z \backslash i)$, then $\phi'^i(x) \geq \phi^i(x)$. This property adheres to the definition of Consistency in the SHAP paper.
> >
> > We now introduce a new property specific to our multi-class explanation. AIM goes beyond what has been achieved in L2X, LIME or SHAP by combing the principles of selection and attribution methods. Our ultimate goal is to identify which features the black-box classifier relies on to output a certain prediction $c$ for a given example $x$.
> >
> > Specifically, we define a feature subset $\mathbb{S}$ to be "relevant" to a class $c$ via a property called Faithfulness, which says that if given the input containing only features in $\mathbb{S}$, the black-box classifier can recover the prediction $c$. Formally, given $x_\mathbb{S}$ indicating the input example that contain only features in $\mathbb{S}$,
> > $$\textrm{argmax} \ f(x_\mathbb{S}) = c,$$
> >
> > Notice that the optimal solution $\mathbb{S}$ corresponds to a single binary pattern $z_x$ where $z^i_x = 1$ if feature $i \in \mathbb{S}$ and $0$ otherwise. We thus can use $h_x(z_x) = z_x \odot x$ to approximate $x_{\mathbb{S}}$ where the excluded features are masked by zeros. In connection with the linear model $g$, we formalize the property of Faithfulness as follows:
> >
> > *iv) Property 4: Faithfulness*
> >
> > Given a binary pattern $z_x$ and the corresponding subset $\mathbb{S} := \\{i \mid  z^{i}_{x} = 1\\}$, if $\textrm{argmax} f(x_\mathbb{S}) = \textrm{argmax} f(h_x(z_x)) = c$, then
> > $$g_c(z_x) > g_j(z_x), \forall j \in \\{1,...,C\\} \backslash \\{c\\}.$$
> >
> > Optimizing our first objective $\mathcal{L_1}$  encourages AIM to converge to attribution scores $\phi_{x}^{i,j} = W_{x}^{i,j}$ that satisfy all the properties discussed above, consequently forcing our local approximation model $W^T_x z_z$ to mimic the correct prediction of $f(h_x(z_x))$ effectively.
> >
> >
> > Reference: Molnar, C. (2018). A guide for making black box models explainable. https://christophm.github.io/interpretable-ml-book/shap.html.

---

> > > ### Author Response · Authors · 2022-12-05
> > > **Properties of AIM attribution scores**
> > >
> > > **Q2. What it should be to get "better" scores (whatever that means), what the the training routine pushes it towards at optimality, or what properties the resulting attribution scores will follow.**
> > >
> > > A novel contribution of our work lies in the elegant multi-class explanation facility via the matrix $W_x$. The scores derived from a column $c$ of the matrix $W_x$ indicate the relative attributions of each feature to class $c$, which are then used to determine feature rankings. At optimality, the scores should capture the relative importance of features so that the subset $\mathbb{S}$ formed by the top $K$ features with highest scores are faithful to the chosen class. This means we do not justify a high-score feature alone, and the goodness of scores depend on whether the output scores determine the optimal ranking of features that result in the most faithful feature subset.
> > >
> > > We must stress that in this work, we mainly focus on explaining the original prediction on $x$. Concretely, we attempt to search for the optimal subset $\mathbb{S}$ that satisfies:
> > > $$\textrm{argmax} f(x_{\mathbb{S}}) = \textrm{argmax} f(x).$$
> > >
> > > Through extensive evaluation on $6$ different metrics along with comprehensive  qualitative and human assessments, we demonstrate that not only does AIM outperforms on Faithfulness but also a plethora of desirable characteristics of an explanation. Table 1 (Section 4.1) details the metrics used to quantify goodness of an explanation. We summarize these characteristics here:
> > >
> > > 1) **Faithfulness (or local fidelity):** whether the output subset of feature can mimic the black-box original prediction on the full input.
> > >
> > > 2) **Causal Importance:** Positive/Negative $\Delta$ Log-odds metrics quantify the degree of importance of features contained in  $\mathbb{S}$ by examining how much the black-box prediction confidence (i.e., log-odds score) drops when removing selected/unselected features. Specifically, features in $\mathbb{S}$ are considered "causally important" if removing them (positive masking) yields a reduction in the confidence, whereas retaining them only in the input (while removing the other features not in $\mathbb{S}$ - so-called negative masking) does not.
> > >
> > > 3) **Stability:** the behavior that similar inputs should have similar explanations. This is a desired effect from amortized global training.
> > >
> > > 4) **Brevity:** an explanation should be brief or concise to be useful to human users. This is an effect from the regularization term $\Vert W \Vert_{2,1}$.
> > >
> > > 5) **Comprehensibility:** a nice by-product of AIM is the capacity of producing comprehensible features that align with human intuition (See human evaluation results Section 4.4). For texts, a low value on Purity metric says that functional stopwords or punctuation are less likely to be selected.
> > >
> > > **One final remark:**
> > >
> > > Besides text modality, we have done additional experiments showing that AIM is also superior on tabular data (See Appendix E.2) while performs competitively with popular baselines on images (See Appendix E.1). Furthermore, our quantitative results are strongly supported by the qualitative analyses along with a human evaluation and comprehensive ablation studies. Importantly, they all show that our performance is there is little variation in our performance under different initialization seeds and our superior performance remains stable across different hyper-parameter settings. Therefore, our empirical evidence should be justified as a signal to confirm the motivations and intuitions, proving the effectiveness and merits of AIM for practical applications.

---

> > > > ### Author Response · Authors · 2022-12-05
> > > > **Motivation of AIM selector (vs. LIME)**
> > > >
> > > > **Q3: This paper replaces the original weighting function with a learned one, but we don't know what that learned weighting function is. The paper doesn't identify a specific issue with LIME's weighting function. It only says it's heuristic and converges slowly, but that doesn't motivate specific changes (except perhaps amortization).**
> > > >
> > > > We respectfully disagree this comment. The selector is not learned to replace the weighting function/kernel, and the slow convergence of LIME is not the main motivation to develop our AIM selector. In the following, we reconfirm the motivations for improving LIME:
> > > >
> > > > 1) LIME learns the explanation for each data example $x$ independently, hence cannot exploit global information carried in many similar examples in a training set. It is obvious that if two data examples $x$ and $x'$ are similar, their explanations should share some global and common characteristics. This motivates us using an amortized network to offer a learnable explanation matrix $W_x$.
> > > >
> > > > 2) To explain the second motivation, we first recap the technicality of LIME. To avoid confusion, we recall the notations used here and note that they are slightly different from the notations used in the original LIME paper. Given a data example $x \in \mathbb{R}^d$, let
> > > >
> > > > * $z \in \\{0,1\\}^d$ be a binary vector for the interpretable (simplified) representation of $x$ indicating the absence/presence of a feature.
> > > >
> > > > * $h_x(z)$ mapping of a binary vector into the original input space $\mathbb{R}^d$.
> > > >
> > > > According to Section 3.4 in the LIME paper, we sample instances $z$ within the input neighbourhood by drawing nonzero elements of uniformly at random (where the number of such elements is also uniformly sampled). Given a perturbed simplification $z \in \\{0,1\\}^d$, the following term is minimized
> > > >
> > > > $$
> > > > \lambda_{x}\left(x'\right)\left(g(z)-f(h_{x}(z)\right)^{2}=\lambda_{x}\left(x'\right)\left(g(z)-f(x')\right)^{2}
> > > > $$
> > > > where the weight $\lambda_{x}\left(x'\right)=e^{-D(x,x')^{2}/\sigma^{2}}$, implying higher weights for closer $x' = h_x(z)$.
> > > >
> > > > The final optimisation problem of LIME is
> > > > $$
> > > > \sum_{x',z}\lambda_{x}\left(x'\right)\left(g(z)-f(x')\right)^{2}+\Omega(g)
> > > > $$
> > > > where $\Omega(g)$ is the regularization term over coefficients of the linear model $g$ for compactness.
> > > >
> > > > We observe that the uniform sampling of $z$ might produce too dispersed set of $z$ and less concentrated on good features of $x$. To further clarify this, let $d= 200$ and assume $x$ has 7 good features and attributions indexed $1,2,...,7$. When sampling $z$ uniformly, a large number of samples $z$ contain no and few good features. As $d$ gets large, training a linear model $g$ on very few informative samples is not effective. The weight $\lambda_x(x')$ favoring closer $x'$ is designed to reduce this negative impact. The questionable quality of LIME's local neighbourhood among many other issues has been actively discussed, namely in Zhao et al., (2021), Slack et al., (2021) & Situ et al., (2021).
> > > >
> > > > It is worth noting that training a local multi-class explanation module in an amortized framework is not trivial. While using such a weighting function with heuristic sampling may be sufficient for LIME, it is not ideal for our purposes.
> > > >
> > > > AIM employs the Selector to balance between exploration and exploitation. On one hand, the stochastic sampling process yields diverse examples with different predictions to help Explainer learn feature attributions w.r.t various classes. On the other hand, mutual information maximization encourages the Selector to sample more of the patterns $z_x$ to the subset of features that can approximate the full-input prediction, **which is our main task**. In this respect, heuristic sampling makes it very difficult for us to effectively control for the quality of local samples that would guide the linear model to focus on the features truly explain the original prediction.
> > > >
> > > > The ablation studies in Appendix D confirm our intuition. It shows how the elegant cooperation of the selector and explainer can offer compact and meaningful features faithfully reflecting the black-box model. Moreover, the superior performance of our AIM to LIME and L2X in all reported measures again consolidates our motivations.

---

> > > > > ### Author Response · Authors · 2022-12-05
> > > > > **Ablation Study & LIME Specifics**
> > > > >
> > > > > **Q5. In the ablation study, training the explainer only is odd because then you have a frozen + random selector; it would make more sense to fix the selector distribution to something sensible like the LIME distribution. On the other hand, training the selector only is odd because L1 is meaningless with random attribution scores.**
> > > > >
> > > > > There seems to be a misunderstanding here. The ablation study in Appendix D aims to verify the importance of each learnable component Explainer and Selector in our framework. We implemented two variant "Training explainer only" and "Training selector only" that borrow the training routine of LIME and L2X respectively. More specifically,
> > > > >
> > > > > * "Training explainer only" removes the learnable Selector and trains Explainer according to LIME's heuristic sampling mechanism. As pointed out above, heuristic sampling is insufficient to train our multi-class module since it is difficult to control the quality of local samples that can focus on good features that can approximate the full input.
> > > > >
> > > > > * "Training selector only" removes the Explainer and trains Selector on $\mathcal{L}_2$ - mutual information maximization term. Similar to L2X, the rankings of features are then determined based on logit values of the Selector's distribution.
> > > > >
> > > > > As reported, these two variants yields a degradation in performance, whose results approximate the main ones reported for LIME and L2X. Jointly training allows each component to focus on optimizing its own tasks most effectively and this experiment support our intuition.
> > > > >
> > > > > **Q6: K is not required, it's an optional step equivalent to fitting the linear model with a Lasso penalty.**
> > > > >
> > > > > For texts, we follow the authors' suggestions in the LIME paper to set a limit on the number of features to encourage interpretability (mentioned in Section 3.4 in LIME paper). Though it is presented in LIME package as an optional argument, the LIME authors must have had some rationale when introducing it as a design choice. Meanwhile, the setup of AIM or FI does not consider or suggest any possibly suitable choice of $K$ during training. However, we agree that the statement "require specifying K" might be viewed as misleading. We noted and will adjust our word choice in the revised version.
> > > > >
> > > > > We further would like to add some details about our implementation: we in fact did not use LASSO/Lars path for feature selection. We set the feature selection method to be ``highest_weights" [(see here)](https://tinyurl.com/2cdzb8x2). It is equivalent to fitting a Ridge regression model and selecting the features with highest value of absolute weight $\times$ original data point. Next, another Ridge regression (with stronger regularization) is fitted to optimize LIME's loss objective. Both models are trained on the same set of local samples and weights, except the latter is only trained on a subset of selected features.
> > > > >
> > > > > Not specifying $K$ is equivalent to skipping the feature selection step and optimize LIME on all features. However, since both steps share the same mechanism, we believe that the faithfulness of features (in our implementation) should not vary significantly between specifying K or not.  To verify our hypothesis, we proceed to run another version of LIME that does not involve $K$ for feature selection. We report the performance of this new version on all datasets, in comparison to our reported results for LIME (train on $K=10$).  The results confirms our hypothesis, and feature selection step perhaps help improve the predictive performance of the linear model or reduce computation costs in practice.
> > > > >
> > > > > |                                             | **IMDB**        |                       | **HateXplain**  |                       | **AG News**     |                       |
> > > > > |---------------------------------------------|-----------------|-----------------------|-----------------|-----------------------|-----------------|-----------------------|
> > > > > | Method                                      | Train on $K$ = 10 | Train on all features | Train on $K$ = 10 | Train on all features | Train on $K$ = 10 | Train on all features |
> > > > > | **Purity (\%) $\downarrow$**                | 36.55$\pm$0.13  | 32.45$\pm$0.05        | 37.73$\pm$0.19  | 33.89$\pm$0.30        | 29.15$\pm$0.08  | 28.12$\pm$0.06        |
> > > > > | **Brevity $\downarrow$**                    | 7.73$\pm$0.15   | 6.54$\pm$0.03         | 7.59$\pm$0.10   | 7.44$\pm$0.04         | 8.76$\pm$0.12   | 8.44$\pm$0.03         |
> > > > > | **Faithfulness (\%) $\uparrow$**            | 79.00$\pm$0.17  | 67.37$\pm$1.07        | 80.56$\pm$0.17  | 79.92$\pm$0.99        | 86.64$\pm$0.06  | 85.18$\pm$0.48        |
> > > > > | **Positive $\Delta$ log-odds $\uparrow$**   | 2.25$\pm$0.18   | 1.07$\pm$0.19         | 3.18$\pm$0.11   | 3.33$\pm$0.07         | 1.38$\pm$0.17   | 1.57$\pm$0.05         |
> > > > > | **Negative $\Delta$ log-odds $\downarrow$** | 5.74$\pm$0.06   | 6.56$\pm$0.07         | 1.07$\pm$0.13   | 1.05$\pm$0.11         | 0.60$\pm$0.13   | 0.77$\pm$0.01         |

---

> ### Comment · Reviewer_qCbz · 2022-12-05
> **Response**
>
> Thanks for engaging so actively in the discussion process. I appreciate your sincere efforts in answering my questions, but after reading your last set of responses I can confirm that I won’t be adjusting my score. Overall, this discussion has not alleviated my main concern that the proposed method lacks a clear theoretical justification. Regarding the relationship with existing properties for SHAP scores, the properties you described are either very different from the original ones or lack the necessary proof to claim that they hold. And the reasons described for requiring a learned subset distribution ultimately sound either 1) related to reducing computation (see computational infeasibility of SHAP, diffuse sampling in LIME), or 2) vague reasons that don’t make sense to me (like contextual dependence between words, focusing on meaningful feature combinations). However, that’s just my judgement and it’s possible that the other reviewers will view things differently.
>
> A couple closing thoughts on your last response, which may be helpful should you further revise the paper:
>
> > “However, this is different from the setting of LIME and not used in practice.”
>
> The kernel you mentioned isn’t used in the LIME package, but it’s used by the SHAP package (see [here](https://github.com/slundberg/shap/blob/master/shap/explainers/_kernel.py)). It’s true that Lundberg & Lee (2017) didn’t discuss the tightness of this approximation, and unfortunately many users likely don’t use enough samples to get accurate estimates (similar to LIME). However, several papers have since revisited the approximation topic to understand the rate of convergence, determine how many samples are needed, improve the convergence speed, and offer alternative approximations (even using amortization). Overall, there’s enough work on estimating SHAP values that I’m not sure they can be ruled out simply due to computational cost.
>
> > “Concretely, for text data, the importance and influence of a word are contextually dependent on other words, meaning that a specific word when combining with other relevant words might be more important. The purpose of text explainers is to select a bag of words that well explains and characterises the predictions of text classifiers.”
>
> It seems inconsistent to criticize SHAP by claiming that the point of text explainers is to select a bag of words, given that you also suggest using AIM by only using the attribution scores. Furthermore, regarding the contextual independence topic, it would be inaccurate to imply that LIME/SHAP don’t consider interactions between words - they do, that’s precisely why they make predictions with many subsets.
>
> > “It might be arguable that the above formulation considers all possible combinations of an arbitrary subset $S$ and $i$, hence also containing meaningful combinations with $i$. However, the number of meaningful combinations are significantly smaller than arbitrary combinations. Therefore, it stands on reason to say that $\phi^i$ does not focus and characterize meaningful combinations.”
>
> It’s not clear what you mean by a “meaningful” combination. Checking many subsets is simply how SHAP/LIME determine which features impact the prediction, and it’s not clear that this would bias the attributions in any negative way. And by referring to the number of feature combinations, it seems that you’re referring to a computational challenge rather than a theoretical motivation for changing the subset distribution (which still seems to be lacking).
>
> > “Regarding L2X, Theorem 1 is a classical result in variational inference. However, solving (1) or (2) in that paper directly is impossible. We need to maximize a lower bound instead. Particularly, the quality of the solution depends on the complexity of the deep net used to define $Q_S(Y \mid x_s)$. The stochastic optimization process certainly introduces more variance to the training process and final solution.”
>
> Yes, L2X is based on a difficult optimization objective and it’s unlikely to always arrive at the optimal solution in practice. But how about this method (AIM), is it any different? It also relies on stochastic training, as well as an objective that’s more complex, so it’s likely to produce variability across runs as well. And unlike L2X, we don’t actually know what it’s approximating.

---

> > ### Comment · Reviewer_qCbz · 2022-12-05
> > **Response (cont.)**
> >
> > > Property 1: Local accuracy
> >
> > 1. This (approximate) property doesn’t involve a summation of all feature scores, so it’s not really the same thing. Since AIM provides feature attributions, it’s worth thinking about properties that apply to those individual scores rather than one specific set of them.
> >
> > 2. You result seems to imply that the solution after training corresponds to a single binary pattern. Is it proved that this would be achieved at optimality, or was it verified empirically? And if this was true, wouldn’t it imply that attribution scores could be split arbitrarily among the features in $S$ so that $g_c(z)$ achieves the correct output? That would suggest the potential for large variability across runs, which would be misleading to users.
> >
> > > Property 2: Missingness
> >
> > The missingness property does not say that missing features get zero score. It says that if a feature’s inclusion does not affect the prediction, it should have an attribution equal to zero. So this version of the property is disconnected from the original one.
> >
> > > Property 3: Consistency
> >
> > The approximate equality $f_x(z) - f_x(z \setminus i) \approx g_c(z) - g_c(z \setminus i)$ seems to come out of nowhere, why would this be the case under AIM? Can you prove that this should hold at optimality, or verify that it holds in practice? The subsequent result seems to depend on this.
> >
> > > “We observe that the uniform sampling of $z$ might produce too dispersed set of $z$ and less concentrated on good features of $x$.”
> >
> > Firstly, you don’t have to sample uniformly and then apply weights; you can sample subsets with probability proportional to the weighting kernel. Secondly, is there a specific issue with dispersion of samples besides convergence speed? For example, SHAP also involves checking every subset, but that’s necessary to satisfy its axioms, so it doesn’t seem harmful except from a computational perspective.
> >
> > > “LIME learns the explanation for each data example $x$ independently, hence cannot exploit global information carried in many similar examples in a training set.”
> >
> > It’s not obvious that you need a learned model to yield similar explanations for similar inputs. If perturbing the features for two inputs affects the predictions in a similar fashion, then LIME/SHAP would recover similar attribution scores automatically. Even after your clarification, the point of amortization seems to be accelerating computation. And I don’t see what the issue is with acknowledging this, other papers are explicit in justifying amortization from a computational perspective (see CXPlain, FastSHAP).
> >
> > > “AIM employs the Selector to balance between exploration and exploitation. On one hand, the stochastic sampling process yields diverse examples with different predictions to help Explainer learn feature attributions w.r.t various classes.”
> >
> > This seems to contradict earlier claims regarding there being a specific binary pattern corresponding to each input. I agree that if you stochastically sample multiple subsets for each input, you have the chance to recover somewhat meaningful attribution scores for each feature; but if the pattern is deterministic (always selecting the “relevant” set of features), couldn’t the scores be split arbitrarily?
> >
> > > ""Training explainer only" removes the learnable Selector and trains Explainer according to LIME's heuristic sampling mechanism."
> >
> > Thanks for clarifying this, that version of the ablation makes sense. Too bad the results aren’t good, it would have been interesting to see an amortized version of LIME work well.

---

### Author Response · Authors · 2022-12-07
**Clarification of AIM motivation (1/4)**

We thank the reviewers for the useful feedback. We kindly provide our responses to recent questions from Reviewer qCbz. We hope to finally clarify our motivation behind AIM and provide a consistent view about our contribution.

**The kernel you mentioned isn’t used in the LIME package, but it’s used by the SHAP package.**

We are aware that the kernel function mentioned in Theorem 2 (SHAP paper) is used to recover the Shapley values, thus not used in LIME package. As mentioned in LIME paper (Section 3.4), LIME specifically uses the exponential kernel function given as

$$\lambda_{x}\left(x'\right)=e^{-D(x,x')^{2}/\sigma^{2}}$$

Their code implementation also confirms our understanding [(here)](https://github.com/marcotcr/lime/blob/03a315cfbfffeabbd12e252c5a82624a0f0e222f/lime/lime_text.py#L351-L353). Furthermore, the function is used for texts, images and tabular data (yet with different distance metrics $D$).

```
def kernel(d, kernel_width):
    return np.sqrt(np.exp(-(d ** 2) / kernel_width ** 2))
```

We only want to emphasize that LIME actually does not satisfy three properties mentioned in the SHAP paper if using the above kernel function.

**It’s not clear what you mean by a “meaningful” combination. Checking many subsets is simply how SHAP/LIME determine which features impact the prediction, and it’s not clear that this would bias the attributions in any negative way.**

To clarify what we mean by a "meaningful" combination, let's take a specific example on texts. Suppose we have the following positive movie review (assuming the classifier predicts correctly):

*the movie left a good impression on us, however due to a personal business, we left during the middle.*

This sentence is tricky in two ways: (1) the first half conveys positive meaning while the second half, if judged alone, conveys the opposite sentiment; (2) the word $left$ has two meanings in this context.

It is clear that the word $left$ when combining with other words carries different meanings, hence we cannot justify the score of this word alone. The combination $\\{left, good, impression\\}$ is a good combination because it conveys positive meaning, while the combination $\\{we, left, during \\}$ is an inappropriate combination in this context.

Shapley scores might have a limitation in this case. When computing the Shapley score of $left$, beside the good combination $\\{movie, left, good, impression\\}$, it also considers all possible combinations like $\\{we, left, during \\}$, $\\{we, left\\}$, $\\{left, middle\\}$, $\\{left, the\\}$, and etc. This is certainly problematic because many combinations convey a negative meaning or are even meaningless. More specifically, when computing the attribution score for the word $left$, Shapley score involves taking the weighted average across all subsets that do not include the word $left$. The score is dominated by a large number of meaningless combinations, including those conveying the opposite meaning.  Therefore, the Shapley score of $left$ cannot focus on a good combination $\\{left, good, impression\\}$ as it should.

**It seems inconsistent to criticize SHAP by claiming that the point of text explainers is to select a bag of words, given that you also suggest using AIM by only using the attribution scores. Furthermore, regarding the contextual independence topic, it would be inaccurate to imply that LIME/SHAP don’t consider interactions between words - they do, that’s precisely why they make predictions with many subsets.**

We indeed do not criticize SHAP. In contrast, we acknowledge and believe that SHAP and LIME are two most important and pioneering works in explainable machine learning/AI, which have been widely appreciated and used in many researches and application domains.

We only want to point out one possible limitation of the Shapley values in some contexts, hence devising a new approach strictly satisfying three properties of Shapley values is not a the only way to have a useful explanation system. Moreover, we do not state that LIME/SHAP don’t consider interactions between words. What we meant was the effect of meaningful combinations may be downplayed by a significantly larger number of arbitrary combinations.

Specifically, our AIM and L2X have the selector network trained by maximizing the mutual information although the purpose is different. For L2X, the selector network directly offers explanations in the form of top $K$ features, with its score updated by assessing the faithfulness from the currently given combinations. For our AIM, the selector presents candidates to the explainer that gradually later in training concentrate more on the meaningful combinations. The explainer thus can benefit from diverse combinations to learn attribution scores while later focus on combinations faithful to the original prediction. Our AIM hence can more effectively take into account the interaction of words for offering faithful and useful explanations.

---

> ### Author Response · Authors · 2022-12-07
> **Clarification of AIM motivation (2/4)**
>
> **And by referring to the number of feature combinations, it seems that you’re referring to a computational challenge rather than a theoretical motivation for changing the subset distribution (which still seems to be lacking).**
>
> We just want to say that by definition, when computing the Shapley value of a feature $i$, we consider all possible combinations (i.e., $2^{d-1}$) of this feature and arbitrary subsets of other features. As mentioned before, most of combinations are not good or even meaningless. Therefore, Shapley value as it is computed is more relevant to the importance of the feature $i$ alone. This possibly makes sense for the context of game theory because we wish to consider what changes to the game if we involve and do not involve the party $i$. Also, in the game context, we do not consider how party $i$ combines with other parties to form a larger party. However, it seems that we need to rethink the application of Shapley values for some specific data such as text. As evidenced later in this thread, Kernel SHAP and Deep SHAP cannot perform well on text data.
>
> Moreover, when computing Shapley values, we need to take average of $2^{d-1}$ terms. It is certainly impossible for quite a large $d$ (e.g., $d=100$). Therefore, the Shapley values should be approximated using sampling technique or solving the linear regression as in Theorem 2 in the SHAP paper. Sampling technique certainly has a large variance due to $2^{d-1}$. For solving the linear regression, we can use either its closed-form with the infeasible complexity $2^d + d^3$ or stochastic gradient descent.  The former is certainly infeasible, while the later requires extreme number of iterations to get converged and if we set a certain number of iterations, we cannot know how far we are from the optimal one. This means that even the practical solutions of SHAP namely Kernel SHAP and Deep SHAP do not satisfy the three desirable properties due to a possibly large gap to the destination.
>
> ### **Our key messages**
>
> First, LIME does not satisfy the three SHAP properties and there is no theoretical guaranty or characteristic. However, it is still very famous and widely used. Our AIM shares the same characteristics as LIME, but adds some more innovative bits to further improve LIME. It is unclear to us why you support LIME but reject our AIM due to the lack of theoretical guaranty or characteristic. The same argument can be applied to Kernel SHAP and Deep SHAP because the ways to compute them cannot ensure theoretical guaranty or characteristic.
>
> Second, we reason that even if some approach can compute the Shapley values exactly, it still suffers from limitations. Therefore, devising new approaches totally based on Shapley values is not the only way to proceed in this problem.

---

> > ### Author Response · Authors · 2022-12-07
> > **Clarification of AIM motivation (3/4)**
> >
> > To further back up our arguments, we run more experiments to compare LIME, Kernel SHAP, Deep SHAP, and AIM on the text datasets. To make it fair, when implementing KernelSHAP, we adopt LIME implementation, using the same setting and only replacing the exponential weighting kernel of LIME with Shapely kernel given in Theorem 2 (SHAP paper). To compare them fairly with AIM, we train LIME and KernelSHAP without any limit on $K$. Due to the limit of time and complexity in adapting SHAP's codes to non-Transformers model, we only run Deep SHAP on IMDB dataset.
> >
> > | **IMDB**                                |    **LIME**    | **Kernel SHAP** |       **AIM**      | **Deep SHAP** |
> > |-----------------------------------------|:--------------:|:---------------:|:------------------:|:-------------:|
> > | Purity (\%) $\downarrow$                | 32.45$\pm$0.05 |  31.12$\pm$0.07 |  **8.22$\pm$0.20** |     15.24     |
> > | Brevity $\downarrow$                    |  6.54$\pm$0.03 |  6.42$\pm$0.04  |  **2.48$\pm$0.01** |     2.67    |
> > | Faithfulness (\%) $\uparrow$            | 67.37$\pm$1.07 |  67.20$\pm$1.00 | **99.62$\pm$0.02** |     51.76     |
> > | Positive $\Delta$ log-odds $\uparrow$   |  1.07$\pm$0.19 |  1.06$\pm$0.02  |  **7.53$\pm$0.03** |     0.90    |
> > | Negative $\Delta$ log-odds $\downarrow$ |  6.56$\pm$0.07 |  6.77$\pm$0.06  | **-0.20$\pm$0.05** |     1.21     |
> > | **HateXPlain**                          |                |                 |                    |               |
> > | Purity (\%) $\downarrow$                | 33.89$\pm$0.30 |  33.85$\pm$0.09 | **19.78$\pm$2.54** |       -       |
> > | Brevity $\downarrow$                    |  7.44$\pm$0.04 |  7.37$\pm$0.02  |  **3.88$\pm$0.21** |       -       |
> > | Faithfulness (\%) $\uparrow$            | 79.92$\pm$0.99 |  77.81$\pm$0.41 | **92.98$\pm$1.17** |       -       |
> > | Positive $\Delta$ log-odds $\uparrow$   |  3.33$\pm$0.07 |  3.01$\pm$0.04  |  **4.98$\pm$0.25** |       -       |
> > | Negative $\Delta$ log-odds $\downarrow$ |  1.05$\pm$0.11 |  1.36$\pm$0.08  | **-1.40$\pm$0.16** |       -       |
> > | **AG News**                             |                |                 |                    |               |
> > | Purity (\%) $\downarrow$                | 28.12$\pm$0.06 |  28.13$\pm$0.05 |  **3.83$\pm$0.05** |       -       |
> > | Brevity $\downarrow$                    |  8.44$\pm$0.03 |  8.45$\pm$0.01  |  **3.39$\pm$0.00** |       -       |
> > | Faithfulness (\%) $\uparrow$            | 85.18$\pm$0.48 |  85.07$\pm$0.89 | **97.92$\pm$0.05** |       -       |
> > | Positive $\Delta$ log-odds $\uparrow$   |  1.57$\pm$0.05 |  1.50$\pm$0.05  |  **7.14$\pm$0.01** |       -       |
> > | Negative $\Delta$ log-odds $\downarrow$ |  0.77$\pm$0.01 |  0.63$\pm$0.12  | **-1.09$\pm$0.02** |       -       |
> >
> > It can be observed that LIME using its own kernel  function, yet not proven to satisfy three SHAP properties, works better than or at lest comparable to Kernel SHAP claimed to have theoretical guaranty. Deep SHAP or Kernel SHAP with the aim to recover Shapley values cannot work satisfactorily on text data which supports our claim about its limitation. Finally, our AIM significantly outperforms Kernel SHAP which demonstrates our claim about its ability in considering the interactions and combinations of words.
> >
> > **Even after your clarification, the point of amortization seems to be accelerating computation. And I don’t see what the issue is with acknowledging this, other papers are explicit in justifying amortization from a computational perspective (see CXPlain, FastSHAP).**
> >
> > We acknowledge that amortization helps accelerate computation. We never deny this fact and we in fact discussed this weakness of LIME in the paper. However, as discussed thoroughly above, this is not the main motivation to propose a learnable Selector. We are onto tackling more subtle issues related to the quality of local samples: what is exactly about the nature of data that hinders convergence. We have demonstrated the usefulness of the selector and amortization via comprehensive experiments and ablation studies.

---

> > > ### Author Response · Authors · 2022-12-07
> > > **Clarification of AIM motivation (4/4)**
> > >
> > > **It’s not obvious that you need a learned model to yield similar explanations for similar inputs. If perturbing the features for two inputs affects the predictions in a similar fashion, then LIME/SHAP would recover similar attribution scores automatically.**
> > >
> > > In the paper, we refer to the property that similar inputs should have explanations as Stability. For texts specifically, the notion of similarity here also refers to the contextual and/or semantic similarity, besides exact match of words. We can have two sentences with the same meaning but with only a few words overlapping.  Specifically, two differences can share the same linguistic pattern , but the pattern is embedded in two sentences at different word locations.
> > >
> > > Given two independent inputs sharing the same linguistic pattern, the black-box model might behave and predict differently in local distributions around two inputs. We are thus not convinced that merely perturbing each input independently and randomly can guarantee that they affect the black-box predictions in a similar fashion given the fact that. It is evident that it is hard for instance-specific approaches like LIME and original SHAP to take advantages of learning global linguistic patterns carried in a text corpus.
> > >
> > > We here expect global training can help the model learn the underlying linguistic patterns that can capture semantically similarity between two inputs. In the context of text explanations, the subsets of selected important words are expected to overlap in large quantities for two similar documents. We evaluate explanation stability through the measure *Intersection over Union* (IoU) (See Appendix B.4 for details).
> > >
> > > On a related note, we summarize how we compute similarity here. Given an example $x$ in the test set, we first search for the nearest neighbors $\mathcal{N}(x)$. The neighboring documents are defined to (1) have the same (black-box predicted) label and (2) be either lexically or semantically similar. We adopt the ratio of overlapping tokens as a proxy metric for lexical similarity. Semantic similarity is measured via cosine similarity of their BERT representations, obtained by summing over the token representations of the last hidden state produced by a pre-trained BERT model.
> > >
> > > **This seems to contradict earlier claims regarding there being a specific binary pattern corresponding to each input.**
> > >
> > > What was precisely stated in our response is that our training routine encourages AIM to focus more on good set of features, which are faithful to the original prediction on input $x$. Such patterns do not arrive immediately. We here recap the details of our the training routine to clarify how it happens.
> > >
> > > Let $c$ denote the original prediction. At first, we initialize the Selector randomly i.e., $\textrm{Multi-Bernoulli}(\pi_x)$ close to a uniform distribution. The sampled binary patterns are diverse and the predicted class $f(z_x \odot x)$ varies. This stage serves the purpose of exploration.
> > >
> > >  Gradually, the Selector tends to choose more relevant features to the class $c$, and the predicted class tends to be more stable at $c$, making the corresponding logit values stable as well. We note that for the column $j = c$, any feature can be offered by the Selector and increased to a higher logit, but the more relevant features to $c$ and their corresponding elements are more likely to be incremented. As a result, the magnitude of elements in the column $j = c$ corresponding to more relevant features tends to be higher. Now recall our third term group norm $||W_x||_{2,1}$: minimizing it creates the sparsity every column $j$, hence encouraging the Selector and Explainer to more agree and focus on the more compact set of relevant features. We empirically investigate the values in $\pi_x$ and observe that the value of $\pi_x^i$ is high for relevant features while staying low for irrelevant features w.r.t the original prediction.
> > >
> > > **If the pattern is deterministic (always selecting the “relevant” set of features), couldn’t the scores be split arbitrarily?**
> > >
> > > Could you please clarify this question? We are not sure how realistic it is for one to deterministically find which patterns are relevant to a black-box prediction. Note that what a machine learning model views important may be obvious from human perspective.

---

### Decision · Program_Chairs · 2023-01-20

**Decision:**

Accept: poster

**Justification For Why Not Higher Score:**

Lack of theoretical analysis.

**Justification For Why Not Lower Score:**

Convincing experimental evaluation, including user study.

**Metareview: Summary, Strengths And Weaknesses:**

The paper proposes a framework for local explanations combining the advantages of attribution-based and selection-based methods, learning local explanations for multiple classes simultaneously.

As the scores are borderline, the reviewers and I had a 1.5-hour discussion over Zoom to discuss the merits of the paper, its drawbacks and whether it is ready for publication at this stage.

The general opinion is that the method has convincingly demonstrated its performance in different scenarios, and could be of use to the ICLR community. I especially appreciated the user study in Appendix F and suggest that a summary of it be given space in the main paper.
The metrics used in the evaluation are reasonable, as a whole, but a graph shown the faithfulness as K is varied (say between 5 and 25) is absolutely necessary for the experimental evaluation to be fully convincing, and should be included in the main paper. Table 11 does include values for different K, so the authors should run experiments for more values, and show the numbers in a graph instead of a table to make it more reader-friendly. The authors should also seriously consider the inclusion of more baselines - see detailed reviewer comments.

In terms of novelty, the use of distribution over subsets of input features was seen as interesting by all reviewers. However, none of us fully understood equation (2), specifically if \tilde{z_x} is a fixed point or a distribution. It it's fixed, then eqn (2) does not incentivize different "attribution" scores. This should be clarified. However, the isight that learning distributions over subsets leads to improvements in faithfulness and the other explainability scores is generally useful to the community beyond this particular method.

The clear, incontestable weakness of the paper is the lack of a precise understanding of why the method outperforms contenders, and without it, the research direction itself cannot be considered complete. This is, however, a slightly separate question from 'does this paper bring a sufficient contribution to warrant publication?'

Reviewer qCbz has rightly pointed out that a theoretical justification would address this concern and would make the paper considerably more useful to the  community, and avoid problems present with past papers published on explainable AI that lacked theoretical foundations.

It is the case, however, that such a theoretical justification is nontrivial to obtain, might involve an extremely simplified version of the algorithm and operating under strong assumptions. Neither myself, nor the reviewers, see the provision of theoretical guarantees as straightforward or trivial in this situation. The authors should make an attempt at it, but this could result in an altogether different paper.

Ultimately, it is my opinion that the paper brings a sufficient contribution from the practical standpoint to warrant publication, though the lack of understanding of why exactly it works is less than ideal.

In addition to the addressing the suggestions above, the authors should address the writing issues pointed out extensively by reviewer qCbz. Some improvements were included in the updated version, however, some were not addressed, for instance the description of LIME: "LIME thus suffers from high variance due to perturbation randomness (Slack et al., 2020; Situ et al., 2021) where there are more of the perturbed examples behave undesirably, for example, to change the original prediction." - this is not quite accurate as sampling perturbations is part of LIME.

**Note From Pc:**

if the above contains the word "oral" or "spotlight" please see: "oral" presentation means -> notable-top-5% and "spotlight" means -> notable-top-25%. As stated in our emails, we are disassociating presentation type from AC recommendations

**Summary Of Ac-Reviewer Meeting:**

The points were included in the review above. The meeting occurred on Friday, Dec 9, 22.